# Solid-State NMR Techniques for the Structural Characterization of Cyclic Aggregates Based on Borane–Phosphane Frustrated Lewis Pairs

**DOI:** 10.3390/molecules25061400

**Published:** 2020-03-19

**Authors:** Robert Knitsch, Melanie Brinkkötter, Thomas Wiegand, Gerald Kehr, Gerhard Erker, Michael Ryan Hansen, Hellmut Eckert

**Affiliations:** 1Institut für Physikalische Chemie, WWU Münster, 48149 Münster, Germany; r.knitsch@uni-muenster.de (R.K.); melanie.siedow@web.de (M.B.); mhansen@uni-muenster.de (M.R.H.); 2Laboratorium für Physikalische Chemie, ETH Zürich, 8093 Zürich, Switzerland; thomas.wiegand@phys.chem.ethz.ch; 3Organisch-Chemisches Institut, WWU Münster, 48149 Münster, Germany; kehrald@uni-muenster.de (G.K.); erker@uni-muenster.de (G.E.); 4Instituto de Física de Sao Carlos, Universidad de Sao Paulo, Sao Carlos SP 13566-590, Brazil

**Keywords:** Frustrated Lewis pairs, aggregation, solid-state NMR, dipolar spectroscopy, internuclear distance measurement

## Abstract

Modern solid-state NMR techniques offer a wide range of opportunities for the structural characterization of frustrated Lewis pairs (FLPs), their aggregates, and the products of cooperative addition reactions at their two Lewis centers. This information is extremely valuable for materials that elude structural characterization by X-ray diffraction because of their nanocrystalline or amorphous character, (pseudo-)polymorphism, or other types of disordering phenomena inherent in the solid state. Aside from simple chemical shift measurements using single-pulse or cross-polarization/magic-angle spinning NMR detection techniques, the availability of advanced multidimensional and double-resonance NMR methods greatly deepened the informational content of these experiments. In particular, methods quantifying the magnetic dipole–dipole interaction strengths and indirect spin–spin interactions prove useful for the measurement of intermolecular association, connectivity, assessment of FLP–ligand distributions, and the stereochemistry of adducts. The present review illustrates several important solid-state NMR methods with some insightful applications to open questions in FLP chemistry, with a particular focus on supramolecular associates.

## 1. Introduction

During the past decade, borane–phosphane frustrated Lewis pairs (FLPs) attracted great interest as metal-free systems for a large variety of catalytic and stoichiometric chemical transformations [1,2,3,4,5,6,7,8,9,10,11,12,13,14,15,16,17]. Their Lewis centers can be incorporated in the same molecule (intramolecular FLPs) or in two different entities (intermolecular FLPs). In these molecules, both Lewis centers are shielded by sterically demanding groups, which prevent *quenching*, i.e., the formation of a covalent bond between them. This so-called *frustration* phenomenon results in a remarkable reactivity rarely encountered in metal-free organic molecules. The most prominent example of such a reaction is the ability of borane–phosphane (B/P) FLPs to split dihydrogen and to transfer the resulting proton/hydride pair to other organic substrates such as unsaturated carbon bonds [1,2,3,4,5,6,7]. In addition to their ability of splitting dihydrogen, B/P FLPs can give rise to various types of adducts with a large number of small molecules, leading to interesting heterocycles and complex organic structures, which might be valuable building blocks for novel organoborane materials. These small molecules include CO_2_, CO, SO_2_, NO, alkenes, alkynes, carbonyls, isonitriles, and many more, as summarized in Figure 1 [1,7,8,9,10,11,12,13,14,15,16,17]. Beginning in 2010, we developed a comprehensive research program exploring the relationship between structure and chemical activity of these systems, using advanced solid-state NMR techniques [18,19,20]. These techniques were not only found to assist in the structural characterization of disordered solids that elude single-crystal diffraction, but to provide suitable anisotropic observables (magnetic shielding, direct and indirect spin–spin coupling, and nuclear electric quadrupolar coupling tensors) that are able to characterize the centers individually, including the extent of intramolecular electronic interactions between them [20,21,22]. As such, NMR spectroscopy was helpful to predict the *degree of frustration* of such systems, which relates to their reactivity. In addition, NMR methods proved to be instrumental in characterizing reaction pathways and intermediates associated with FLP chemistry, as well as for the stereochemical analysis of the final products obtained [12,21,22,23,24,25]. As discussed in previous reviews [18,19], NMR spectroscopic parameters allow a distinction between classical and cooperative adducts, the identification and quantification of different possible isomers and epimers, and the description of intermolecular aggregation and oligomerization effects.

The formation of supramolecular macrocyclic structures by FLPs and their small-molecule adducts was developed further during the past five years, and it forms the focus of the present review. Such supramolecular assemblies are not only interesting in their own right from a sheer esthetics viewpoint, but also because they represent small spin-cluster systems, allowing advanced solid-state NMR methodologies to be tested, validated, and optimized with the help of theoretical calculations and simulations. In the present review, we illustrate how advanced solid-state NMR methodology can be used to obtain essential information on the structural organization of such complex materials, which provide important insights into their reactivity and self-assembling features. The use of solid-state NMR spectroscopy is crucial for these purposes, as the supramolecular arrangements may fall apart into their monomeric constituents upon dissolution. Furthermore, the anisotropic interactions, like dipolar and quadrupolar couplings, which are most intimately linked to structural information, are not accessible in the liquid state. Here, we focus on measurements of crystallographically well-characterized materials, thereby laying the foundations for future applications to FLP-based systems in poly-, nano-, or disordered crystalline and amorphous states of matter.

## 2. Solid-State NMR—Basics and Methodology

### 2.1. Fundamental Principles

The basic principles of nuclear magnetic resonance (NMR) were summarized in numerous excellent books and review articles devoted to applications to various fields in solid-state chemistry [26,27,28,29,30]. The method is based on nuclear spin motion, represented by an angular momentum operator *J*.
(1)J=Iℏ
where ℏ is the reduced Planck’s constant, and ***I*** is a dimensionless operator, satisfying angular momentum commutation rules and quantization conditions.
(2)$$\left|I^{2}\right|=I\left(I+1\right)$$
(3)$$\left|I_{z}\right|=m\;\varepsilon \;\left\{I,I-1,\ldots ,-I\right\}.$$

Thus, the nuclear spin angular momentum is orientationally quantized into 2*I* + 1 degenerate states. Here, *I* denotes the nuclear spin quantum number, which is also nucleus-specific and adopts one of the half-integer values 1/2, 3/2, 5/2, 7/2, or 9/2 for stable nuclei (with the exception of a few nuclei with integer spin). Nuclear spin angular momentum entails nuclear magnetism, according to the following relationship:(4)μ=γIℏ
where *γ*, the *gyromagnetic ratio*, is a nuclear specific constant. In NMR, these nuclear magnetic moments *µ* can be detected by applying an external magnetic field, represented by the magnetic flux density *B*_0_. The resulting *Zeeman interaction* lifts the energetic equivalence of orientational states and causes a splitting into 2*I* + 1 individual levels with energies
(5)$$E_{m}=-m\gamma \hbar B_{0}$$
where *m* denotes the orientational quantum numbers, which, in the classical picture, correspond to spin orientations that are partially aligned or counter-aligned with the direction of $B_{0}$. By application of electromagnetic waves satisfying the Bohr condition ΔE=ℏω, allowed transitions between adjacent states (Δm=±1) can be stimulated and observed spectroscopically. The interaction takes place between the nuclear magnetic dipole moments and the oscillating magnetic field of the applied electromagnetic wave. The angular resonance frequency is given by
(6)$$\omega =\gamma B_{0}$$
which is equal to $2\pi \upsilon_{P}=\gamma B_{0}$, where $\upsilon_{P}$ is the *Larmor frequency* with which the nuclei precess in the applied magnetic field. Since the values of *γ* differ greatly for different kinds of nuclei, NMR spectroscopy is intrinsically element-selective as, at a given value of $B_{0}$, different nuclear isotopes have different resonance and precession frequency ranges. With typical values of applied magnetic flux densities between 4.65 and 23.6 T used in NMR, the frequencies lie in the radio-wave region (10–1000 MHz), depending on the value of *γ*. The precession frequencies are measured using pulsed excitation, recording the resulting *free induction decay (FID)*. Fourier transformation then produces the frequency domain signal.

### 2.2. Internal Nuclear Interactions 

While nuclear precession frequencies measured in NMR are usually dominated by the Zeeman interaction, they are additionally influenced by a number of *internal interactions*, whose parameters reflect the details of the local structural environment. The electronic environments around the nuclei produce *magnetic shielding* effects that modify the local fields experienced by the nuclei. These electronic environments also produce *electric field gradients* (EFGs), which can interact with the nuclear electric quadrupole moments of spin >½ nuclei. In addition, the electronic environment can promote internuclear spin–spin coupling by a spin-polarization mechanism (*through-bond coupling*). Finally, even in the absence of bond connectivity, the local magnetic fields are altered by direct (*through-space*) *dipole-dipole coupling*, whose strength is proportional to the inverse cubed internuclear distance. The strengths and spin dynamics of the latter two interactions further depend on whether the interacting nuclei are of the same or of different kinds (homo- versus heteronuclear couplings). Figure 2 summarizes the internal interactions, their most important NMR parameters, and their relevance for structure elucidation. They refer to the common case of diamagnetic solids; additional interactions are present in paramagnetic materials [31]. 

In the solid state, all internal nuclear spin interactions are anisotropic, which means that their effects on the nuclear precession frequency depend on molecular orientation in the magnetic field. Thus, the NMR spectra of polycrystalline samples are significantly broadened, owing to the statistical orientation distribution of the crystallites. Furthermore, the four internal interactions (Figure 2) are usually superimposed, which makes the resulting NMR spectra difficult to analyze. However, as detailed further below, this superposition can be disentangled by special *selective averaging* experiments. This disentanglement is usually essential for linking the spectroscopic details to structural information. The following sections introduce the features of the four internal interaction mechanisms and their link to structural information. Alongside, we describe the experimental strategies that are most commonly used in FLP chemistry for extracting such interactions in a selective fashion.

#### 2.2.1. Magnetic Shielding Effects 

Basic Principles: The externally applied magnetic field *B*_0_ induces magnetic polarization effects in the core and valence shell electronic environment of the nuclei, influencing their nuclear precession frequencies. This anisotropic effect, called *magnetic shielding*, is described by a second-rank tensor, which can be parametrized in terms of the following three parameters: *σ*_iso_ = 1/3(*σ*_33_ + *σ*_22_+ *σ*_11_), which corresponds to the isotropic average value; Δ*σ* = *σ*_33_ − 1/2(*σ*_11_ + *σ*_22_), its anisotropy and *η*_σ_ = (*σ*_22_ − *σ*_11_)/(*σ*_33_ − *σ*_iso_), its asymmetry parameter, that is, the deviation of the tensor from axial symmetry [32]. In the simplest case of an axially symmetric magnetic shielding anisotropy, $\omega_{P}$ is given by
(7)$$\omega_{P}\left(\theta \right)=2\pi \nu_{P}\left(\theta \right)=\gamma B_{0}\left(1-\sigma_{iso}-\frac{1}{3}\Delta \sigma \left(3cos^{2}\theta -1\right)\right),$$
where the angle *θ* specifies the orientation for the axis of the magnetic shielding main component *σ*_33_ relative to the magnetic field direction. Note that, for *θ* = 54.7°, the term $3cos^{2}\theta -1$ is zero, and only the isotropic value is measured. 

Experimental Determination: The anisotropy of $\omega_{P}$ expressed by Equation (7) can also be removed if the time average of the term $3cos^{2}\theta -1$ can be brought to zero. Specifically, this is achieved by the *magic-angle sample spinning* (MAS) technique, where the NMR spectrum is measured while rapidly rotating the sample about an axis inclined by an angle of *β*_m_ = 54.7° relative to the magnetic field direction [33]. This manipulation creates the same average orientation < θ> of 54.7° for all molecules or structural fragments (irrespective of their original orientation), if rotation is sufficiently rapid, leaving the same isotropic average of $\upsilon_{P}$ to be measured for all spins in the same chemical environment. The result is a significant line-narrowing effect allowing the identification of distinct local environments on the basis of their individual isotropic magnetic shielding constants *σ*_iso_. Magnetic shielding values can currently be calculated with high accuracy using suitable quantum chemical methods [34]. Experimentally accessible values, however, are not absolute shielding values (which would require comparisons with bare nuclei) but rather *chemical shifts* measured relative to the precession frequency of a reference compound.
(8)$$\delta_{iso}=\frac{\upsilon_{sample}-\upsilon_{ref}}{\upsilon_{ref}}.$$

If the rotation frequency $\nu_{R}$ does not fulfill the condition $2\pi \nu_{R}$ >> $\gamma B_{0}\;\Delta \sigma$, the narrowed NMR signal is accompanied by *spinning sidebands*, which are spaced from the central signal at integer multiples of ν_R_ (see Figure 3). When quantifying relative ratios of spectroscopically resolved species, the areas of these spinning sidebands must also be taken into account. Furthermore, in the slow-spinning limit ($2\pi \nu_{R}$ << $\gamma B_{0}\;\Delta \sigma$), accurate values of Δ*σ* and *η*_σ_ can be extracted from the intensity profiles, providing additional information regarding the local symmetry of the nuclear species observed [35]. On the other hand, the spinning frequency required for obtaining simple single-peak-per-site spectra increases linearly with *B*_0_, as the spectral dispersion (in Hz) caused by the chemical shift anisotropy is proportional to the applied magnetic field strength. Currently, such experiments are performed at MAS frequencies up to ~110 kHz achieved with commercially available rotors of 0.7 mm diameter, although even higher MAS frequencies using even smaller-sized rotors were reported recently [36,37].

#### 2.2.2. Nuclear Electric Quadrupole Interactions

Basic Principles [38]: For nuclei with a spin quantum number *I* > ½, the charge distribution is not spherically symmetric. This asymmetry can be described by an electric *quadrupole moment* superimposed upon a sphere containing the nuclear charge. Classical physics predicts an interaction of these moments with inhomogeneous electric fields, i.e., electric field gradients $V_{ij}$ (EFGs), present at the nuclear sites. The latter are generated by the asymmetric charge distributions associated with the electrons of the atom bearing the nucleus, as well as atomic coordination and chemical bonding effects. The EFG is a symmetric second-rank tensor, which can be diagonalized in a molecular axis system. The sum of the diagonal elements $V_{xx}+V_{yy}+V_{zz}$ vanishes (Laplace equation) such that the interaction can be described in terms of two parameters, the nuclear electric quadrupolar coupling constant (in frequency units),
(9)$$C_{Q}=\frac{eQV_{zz}}{h},$$
and the asymmetry parameter,
(10)$$\eta_{Q}=\frac{V_{xx}-V_{yy}}{V_{zz}},$$
where 0 ≤ $\eta_{Q}$≤ 1 describes the deviation of the EFG tensor from cylindrical (axial) symmetry. Figure 4 illustrates typical electronic environments in molecular inorganic chemistry and their associated EFG properties. 

Experimental Determination: The quadrupole interaction competes with the Zeeman interaction for quantized spin alignment. For the discussion of NMR spectra, the case of a dominant Zeeman interaction is important where simple perturbation theory can be applied. It produces anisotropic shifts in the Zeeman energy levels, which are proportional to the square, *m*^2^, of the Zeeman orientational quantum number. As a consequence, in the case of very weak quadrupole interactions, the central m=1/2↔ m=−1/2  transition is unaffected and can be analyzed like a signal from spin-1/2 nuclei. In contrast, the other allowed Δ*m* = ±1 transitions (the so-called *satellite transitions*) are anisotropically broadened and may be (in the case of stronger quadrupole couplings) shifted so strongly off-resonance that they are only incompletely excited by the radiofrequency pulses. Stronger quadrupole couplings will also lead to more complicated orientation dependences for the central transition, due to the presence of higher-rank tensors in the quadrupolar Hamiltonian, producing line-broadening effects that scale with the inverse of the strength of the applied magnetic field (*second-order effects*), and that can be only partially averaged out by magic angle spinning [38]. For crystalline compounds with well-defined lattice positions or bonding geometries, often highly characteristic line shapes can be observed, from which $C_{Q}$ and $\eta_{Q}$ can be extracted via line-shape simulation (see Figure 4), using standard program packages such as DMFit [39], SIMPSON [40], or QUEST [41]. In addition, first-principles calculations of EFG tensor values are possible using standard theoretical programs such as GAUSSIAN [42], TURBOMOLE [43], WIEN2k [44], or CASTEP [45].

An improved resolution for quadrupolar nuclei affected by such anisotropic line broadening is available using the multi-quantum spectroscopic (MQMAS) method [46,47,48]. This technique correlates the evolution of an |m⟩↔|−m⟩ coherence with the |½⟩↔ |−½⟩ coherence (the central transition) in a two-dimensional NMR experiment. While the excitation of coherence orders >1 is symmetry-forbidden, this selection rule is relaxed owing to the substantial quadrupolar perturbation of the Zeeman state functions, such that these coherences can be excited in this case. Essentially, the design of the method is to reverse the anisotropic time evolution of the spin system during the time period *t*_1_ during the detection period *t*_2_, leading to an isotropic spectrum after Fourier transformation with respect to the evolution time variable *t*_1_. In the spectroscopic characterization of FLPs, the most important application concerns the ^11^B nuclei, having spin 3/2, whose strong quadrupolar interaction results in broadened line shapes of the type shown in Figure 4b. In samples containing multiple boron species, this produces strongly overlapping MAS NMR signals which make the deconvolution analysis difficult at times. Here, the triple-quantum (TQ) experiment involving the |3/2⟩↔ |−3/2⟩ transition can produce significant improvement in the resolution. Figure 5 shows the most common pulse sequence in use; the first pulse P1 creates triple quantum coherence which is allowed to evolve for the time period *t*_1_, after which it is converted back to zero-quantum coherence by the second pulse P2. Detection follows after converting the zero-quantum to single-quantum coherence, i.e., magnetization. For both pulses P1 and P2, high-radiofrequency amplitudes are required to ensure efficient excitation of the triple quantum coherence, whereas the detection pulse is chosen to be soft, to be limited to singlequantum coherence excitation (creation of transverse magnetization from coherence level zero). For appropriate coherence selection, special phase cycling schemes have to be used. 

Figure 6 shows a typical application result of an FLP octamer discussed in more detail in Section 3.6. This compound features two crystallographically distinct boron sites B1 and B2, whose line shapes are strongly overlapping in the regular MAS NMR spectra. Double Fourier transformation of the two-dimensional (2D) dataset acquired with this pulse sequence results in a two-dimensional map in which the regular MAS NMR spectrum (plotted in Figure 6 along the horizontal dimension) is correlated with an isotropic spectrum (plotted along the vertical dimension). Note the excellent resolution in the isotropic dimension, from which the individual MAS NMR line shapes can also be extracted by simulating the corresponding F1 cross-sections, yielding the parameters $\delta_{iso}^{cs},\;C_{Q},$ and $\eta_{Q}$ (see Figure 6) for both boron sites. 

#### 2.2.3. Indirect Spin–Spin Coupling (“J-coupling”) 

Basic Principles: The electronic environment of the nuclei does not only alter nuclear precession frequencies via the magnetic shielding or electric quadrupole coupling interactions, but it can also mediate internuclear spin–spin interactions. In essence, the spin orientation of an individual nucleus (I) induces a slight spin polarization of the bonding electrons linking this nucleus with another one (S), altering the resonance frequency of the latter. The effect is anisotropic and, thus, requires a tensorial description [49,50]. Under (sufficiently fast) MAS conditions, only the isotropic component is measured, resulting in a characteristic peak splitting, from which the number of bonded neighbors can be inferred. In the simplest case of a pair of two nuclei of spin ½, the resonances of both the I and the S nuclei are then split into doublets, each reflecting the two distinct possible orientational states of the bonded nuclei. In general, the coupling of an observed nucleus S to *n*_j_ equivalent nuclei of spin quantum number *I*_j_ leads to a peak multiplicity given by Π_j_ (2*n_j_I_j_* + 1). In this expression, both homo- and heteronuclear contributions must be considered, where the latter can be removed by radiofrequency decoupling schemes (see Section 3.3). The magnitude of the splitting (in Hz) is field-independent and given by the *spin*–*spin coupling constant*, *J_iso_*. Using this symbol, the indirect spin–spin interaction is commonly called *J-coupling* to differentiate it from the through-space dipolar coupling (see below). If quadrupolar I nuclei exposed to strong EFGs are involved, the splittings become asymmetric and unevenly spaced (see Figure 7). The simulation must then include an additional dipolar coupling parameter *d*, which depends on the quadrupolar coupling strength [51,52]. The principal structural information obtained from the J-coupling is evidence for through-bond connectivity (although, in some special cases, no-bond indirect spin–spin couplings were also observed [53,54]). The value of *J*_iso_ depends not only on the sizes of the nuclear magnetic moments, but also, in particular, on the electron distribution in the chemical bond (bond covalency). For intramolecular borane–phosphane FLPs, the ^31^P–^11^B J-tensors can be calculated with good accuracy using standard density functional theory (DFT) methods [20].

Experimental Determination: Frequently, J-splittings such as those evident in Figure 7 are not experimentally observable under MAS conditions because the line shapes are affected by other broadening mechanisms. In this case, two-dimensional spectroscopic approaches prove instrumental. The simplest method is the J-based heteronuclear resolved experiment illustrated in Figure 8 [55]. The method exploits a time-reversal strategy using spin inversion with 180° pulses; chemical shift evolution during an initial time period *t*_1_/2 is canceled by chemical shift evolution during a second time period *t*_1_/2 following a π pulse; all the while, evolution under J-coupling continues unaffected by the latter. In the heteronuclear (I–S) version, both types of nuclear spins need to be inverted by π pulses to ensure continued evolution. This evolution is stopped by the subsequent π/2 pulses on both channels (producing z-filtering), while the following π/2 pulses create transverse magnetization, to evolve during *t_2_* under the combined effects of the Zeeman interaction and indirect spin–spin couplings. Double Fourier transformation of the 2D dataset then produces a 2D plot in which the regular MAS NMR spectrum (detected during *t*_2_) is correlated with the multiplet (a doublet for a two-spin system), arising from indirect spin–spin coupling monitored during the incremented evolution time *t*_1_.

J-Based Correlation Spectroscopy: Several different NMR experiments were designed to correlate heteronuclear NMR resonances exploiting J-couplings, of which the *insensitive nuclei enhanced by polarization transfer* (INEPT) sequence proved to be well suited for FLP macrocycles offering sufficiently large isotropic J-couplings. The INEPT experiment is widely used as a signal amplification technique in liquid-state NMR, but it is equally effective under MAS conditions [56]. The sequence depicted in Figure 9a begins with a Hahn-echo at the non-observed nucleus. A simultaneous 180° pulse on the observed channel recouples the isotropic indirect dipolar interaction, while the direct dipolar coupling is averaged out by MAS as $\tau_{1}$ typically exceeds several rotor periods. Thus, evolution during $\tau_{1}$ proceeds exclusively due to J-coupling. The optimum timing for maximizing I–S polarization transfer via the two simultaneous 90° pulses on both channels is *τ*_1_ = *τ*_2_ = 1/4*J*. The transferred polarization can either be detected on the insensitive (see Figure 9) or on the more sensitive nucleus, which forms the polarization source (*indirect detection*), resulting in a considerable sensitivity enhancement. In the two-dimensional version, a heteronuclear correlation plot is observed in which covalently bonded nuclei give rise to cross-peaks. Alternatively, J-based heteronuclear correlation spectra can also be obtained via heteronuclear double quantum spectroscopy applied under standard MAS conditions [57] (for more information on double-quantum (DQ) spectroscopy, see Section 2.2.4). In that pulse sequence, given by 90°(S)–(1/2*J*)–90°(I)–*t*_1_–90°(I)–acquire(S, *t*_2_), S–I double quantum coherence is first excited by the pair of 90°(S) and 90°(I) pulses, which must be separated by the time period ½ *J*. This DQ coherence is allowed to evolve for an incremented evolution time *t*_1_, before it is stopped by a second 90° pulse applied to one of the nuclear species (I). This pulse also triggers the detection period, where the regular free induction decay of the S nuclei is being acquired. As the double quantum coherence evolves with the sum of the chemical shifts of the I and S nuclei involved, this simple sequence, equipped with the appropriate phase cycling schemes for coherence selection, can be used under regular MAS conditions for indirect detection of insensitive nuclei, spectral editing via heteronuclear double quantum filtration, and two-dimensional correlation spectroscopy [57]. The analogous homonuclear version is called INADEQUATE (*Incredible Natural Abundance Double Quantum Transfer Experiment*) [58]. Although it enjoys widespread use in other areas of materials science, so far, no such experiments were applied to borane–phosphane systems, presumably as the detection of P---P bond connectivity is usually not an issue in FLP chemistry.

#### 2.2.4. Direct Spin–Spin (Dipolar) Coupling

Basic Principles: Nuclear precession frequencies are further influenced by magnetic dipole–dipole interactions, as the magnetic moments of neighboring spins create local magnetic fields that affect the precession frequencies of the probe nuclei. These depend both on distances and on orientations (defined by the angle of the internuclear distance vector relative to the magnetic field direction). Again, there is a homonuclear contribution describing interactions of the observed nuclei with spins of the same kind and a heteronuclear contribution describing interactions of the observed nuclei with spins of different kinds. Their corresponding Hamiltonians relevant for the NMR line shape in the limit of first-order perturbation theory are given by
(11)$${\hat{H}}_{DD,i,j}^{homo}=-d_{i,j}\;\left(3cos^{2}\theta -1\right)\left[{\hat{I}}_{z,i}\;{\hat{I}}_{z,j}-\frac{1}{2}\left({\hat{I}}_{x,i}\;{\hat{I}}_{x,j}+{\hat{I}}_{y,i}\;{\hat{I}}_{y,j}\right)\right],$$
(12)$${\hat{H}}_{DD,i,j}^{hetero}=-d_{i,j}\;\left[{\hat{I}}_{z,i}\;{\hat{I}}_{z,j}\right]\left(3cos^{2}\theta -1\right)\;\;\;\;\;\mathrm{with}\;\;\;\;d_{i,j}=\frac{\mu_{0}\gamma_{i}\gamma_{j}\hbar^{2}}{4\pi r^{3}}\;,$$
where the operators ${\hat{I}}_{z,i}$, ${\hat{I}}_{x,i}$, and ${\hat{I}}_{y,i}$ and ${\hat{I}}_{z,j}$, ${\hat{I}}_{x,j}$, and ${\hat{I}}_{y,j}$ stand for the cartesian spin angular momentum components of the nuclei *i* and *j*. As indicated by Equation (12), the orientation dependence results in line-broadening effects for polycrystalline samples. The dipolar coupling constant *d*_i,j_ is proportional to the inverse cube of the internuclear distance *r*, providing a straightforward connection to geometric structure. 

Recoupling Heteronuclear Dipole Interactions: The most important experimental technique for measuring heteronuclear dipolar interactions under the high-resolution conditions of MAS NMR is the *rotational echo double resonance* (REDOR) experiment [59,60,61]. As illustrated in Figure 10, MAS causes the dipolar coupling to oscillate during the rotor period *T_R_* leading to its cancellation over a complete rotor cycle. However, if the sign of the dipolar Hamiltonian is inverted by applying a *π*-pulse to the non-observed *I-*spins, this average is non-zero and the interaction is said to be *re-coupled*.

One then measures the normalized difference signal *ΔS/S_o_* = *(S_o_ − S)/S_o_* in the absence (intensity *S_o_*) and the presence (intensity *S*) of the recoupling pulses. The signal of the observe nuclei S is usually detected by a rotor-synchronized Hahn echo, while the dipolar re-coupling is effected by 180° pulses during the rotor period, applied to the I nuclei. By measuring *ΔS/S_o_* as a function of dipolar evolution time *NT_R_*, i.e., the duration of one rotor period multiplied by the number of rotor cycles, the strength of the dipole–dipole coupling can be characterized. Pulse sequence imperfections and finite pulse length effects can be corrected by the compensation scheme shown in Figure 11 (bottom) [62]. In this method, an additional π(I) pulse applied simultaneously with the π(S) refocusing pulse eliminates the re-coupling effect. This “dummy” sequence can be considered a more realistic reference experiment than the standard rotor synchronized echo sequence, even though the inclusion of a third set of measurements tends to reduce overall signal-to-noise ratio and there seems to be a tendency to “over-correct” at longer evolution times [62]. 

In the case of *I* > ½ nuclei, the S{I} REDOR experiment is complicated by the quadrupolar interaction, which produces large off-resonance shifts for the nuclei involved in the non-central Zeeman transitions, which in this case do not resonate with the applied REDOR π recoupling pulses. For such cases, the *rotational echo adiabatic passage double resonance* (REAPDOR) [63,64] technique is very useful. This method replaces the π recoupling pulses by continuous-wave irradiation of the quadrupolar nuclei during a duration of $T_{R}/3$ [65]. As the energy levels of the nuclei in the non-central Zeeman states are modulated by MAS, they can come into resonance temporarily while the radiofrequency field is on for the quadrupolar nucleus, leading to population transfers between these levels during the irradiation period. In this manner, dipolar interactions of the observe nuclei with the quadrupolar nuclei in non-central Zeeman states are being re-coupled as well, enhancing the overall dipolar dephasing effect in a calculable fashion. Approximate analytical expressions for short mixing periods were also given [66].

Recoupling Homonuclear Dipolar Interactions: For detecting homonuclear dipolar couplings under MAS conditions, a frequently utilized approach is the excitation of double-quantum (DQ) coherences. Such coherences involve the combined excitation of two simultaneous *Δm* = 1 transitions for two nuclear spins. This effective *Δm* = 2 transition can only occur if the two nuclear spins are coupled to each other. While both through-space and indirect spin–spin coupling can be exploited for DQ excitation, the former is predominantly utilized to obtain structural information in FLP chemistry. The simplest DQ excitation scheme involves the application of trains of two short 90° pulses (*back-to-back* (*BaBa*) sequence [67,68] (see Figure 12)); however, alternative, more sophisticated excitation schemes based on gamma-encoding are also known [69,70,71]. 

DQ excitation can be exploited by proving spatial proximity between nuclei within distinct, spectroscopically resolved sites [69]. The pulse sequence in this case contains an evolution period *t*_1_, during which the DQ coherence evolves, followed by reconversion to single-quantum (SQ) coherence by a 90° pulse after which the regular MAS NMR signal is monitored during the detection period *t_2_*. Separation of SQ and DQ coherences is easily achieved by appropriate phase cycling. As DQ coherence evolves with the sum of the involved resonance frequencies of both respective spins, the corresponding 2D spectrum can serve to correlate the resonance frequencies of proximal spins, leading to off-diagonal *cross-peaks* between neighboring inequivalent species and diagonal *autocorrelation peaks* for proximal chemically equivalent spins.

The DQ signal intensity depends on both the strength of the magnetic dipole–dipole coupling and the total length of the excitation period. Measurements of such DQ coherence build-up curves can be used for the quantitative characterization of spin systems and distance measurements [68]. A specific variant of this principle is the DRENAR (*dipolar re-coupling effects nuclear alignment reduction*) pulse sequence, which monitors an intensity difference in analogy to the REDOR method [72,73,74,75]. DRENAR measures the decrease of longitudinal magnetization caused by the build-up of DQ intensity. The standard sequence depicted in Figure 13 consists of two consecutive DQ excitation blocks, followed by a 90° detection pulse for the residual magnetization. For the reference signal, the second block is phase-shifted by 90° (C’), leading to inversion of the homonuclear Hamiltonian and, hence, cancellation of the DQ coherence. In the standard DRENAR sequence [72,73], the difference intensity (without and with recoupling) is measured as a function of the excitation time. In the constant-time (CT) variant, the phase shift of the second block relative to the first block is systematically incremented rather than the DQ excitation time [74].

### 2.3. Cross-Polarization

Finally, the magnetic dipole–dipole coupling is also exploited in *cross-polarization* (CP), in which polarization of a high-*γ* abundant spin system (usually protons) is transferred to recipient heteronuclei whose enhanced signal is then detected. To achieve an energy-conserving heteronuclear magnetization transfer, an identical Zeeman splitting for both isotopes involved is required, corresponding to equal precession frequencies of both nuclear species I and S. This is only possible in the doubly rotating frame, with radiofrequency pulses applied to both nuclei [76] under spin-locking conditions, adjusting both radiofrequency amplitudes according to the MAS-modulated Hartmann–Hahn *matching condition* [77] (see Figure 14).
(13)$$\nu_{1,I}\pm n\times \nu_{R}=\nu_{1,S},$$
where *ν*_1,I_ and *ν*_1,S_ are the nutation frequencies of the I and S nuclei, *ν*_R_ is the rotor frequency, and *n* is an integer. A part of the enhanced detection sensitivity arises from the much shorter spin-lattice relaxation times of the ^1^H source nuclei compared to the recipient heteronuclei. In fact, this is the main reason why ^31^P MAS NMR spectra on FLPs are usually obtained via cross-polarization from ^1^H spin reservoirs. The key to the cross-polarization principle is the fact that the abundant-spin system in the spin-locked state is far away from equilibrium conditions and tends toward equilibrium by losing magnetization. While this naturally occurs due to spin-lattice relaxation (time constant $T_{1\rho }$, Figure 14c), the double irradiation experiment under the condition in Equation (12) opens up an alternative cross-relaxation path, via which the magnetization is channeled to the heteronuclei utilizing the magnetic dipole–dipole coupling mechanism. This transfer is characterized by the transfer time constant $T_{CP}$, which depends on the strength of the heteronuclear dipolar coupling. Owing to the interplay of these competing relaxation channels, the achievable signal enhancement will depend on the duration of the double irradiation period (the so-called *contact time,*
$t_{ct}$) which must be optimized experimentally on the sample.

## 3. Results and Discussion

### 3.1. NMR Criteria of FLP Behavior

Specific characteristics of vicinal borane–phosphane frustrated Lewis pairs (B/P FLPs) as determined experimentally and confirmed by DFT calculations were previously reviewed [19,78,79,80]. Both FLP centers are mostly characterized by an “in-between” coordination state between three and four, owing to the partial covalent bond between the Lewis centers, affecting the ^11^B chemical shifts and the ^11^B nuclear electric quadrupolar coupling constants. Figure 15 illustrates that both parameters are well correlated with each other, indicating that the strength of this partial bond dominates the trends in both NMR observables [79]. In addition, these parameters are well-correlated with the B^…^P internuclear distance, which varies between 2.00 Å (strongly interacting) and 2.20 Å (weakly interacting Lewis pairs). As a feature common to all vicinal B/P FLPs studied so far, the principal axis of the ^11^B EFG tensor makes an angle of ~21 ± 3° with the B---P vector [19]. The covalent interaction also manifests itself in ^11^B–^31^P indirect spin–spin couplings with an isotropic coupling constant of up to 60 Hz. ^11^B{^31^P} REDOR experiments further indicate a contribution from a *J*-anisotropy Δ*J* on the order of ~100 Hz.

### 3.2. Dimeric Structures

Molecular aggregation, oligomerization, and polymerization of FLPs are often observed as a consequence of strong covalent intermolecular interactions between the monomers. In the case of the cyclic FLP **1** shown in Figure 16, crystallization produces a simple dimer [25]. A dimeric structure was also obtained for the adduct **2**, which is prepared by reacting the polymeric FLP [Ph_2_P–CH=CH–B(C_6_F_5_)_2_]_n_ with benzaldehyde (the crystal structure **2** will be the subject of a subsequent publication). While the completely amorphous nature of the polymeric material precludes a conventional structural characterization using single crystals, solid-state NMR was able to develop important structural constraints [19,78]. An essential feature of B/P FLP aggregates concerns the structural analysis by REDOR and REAPDOR experiments, which can no longer be analyzed in terms of simple ^11^B–^31^P two-spin interactions. Rather, the multi-spin character of these interactions must be taken into account. This is illustrated in Figure 16b as an example of the ^31^P{^11^B} REAPDOR curves measured for **1** and **2**.

As each of the ^31^P nuclei interacts with two closest ^11^B spins, the existence of different isotopologues must be considered, reflecting the natural abundances of ^11^B and ^10^B nuclei. For compound **1**, these contributions consist of a dominant three-spin component (64.2%, distances 2.08 and 2.75 Å, ^11^B^…31^P^…11^B angle of 90°, blue curve) and two minor two-spin components from the two possible ^11^B–^31^P–^10^B isotopologues (15.9% each), while 4% of the molecules do not contain any ^11^B and, thus, will yield no REAPDOR effect. For compound **2**, the situation is completely analogous, except that the B^…..^P distances are much longer, i.e., 3.94 and 4.27 Å, respectively, with a ^11^B^…31^P^…11^B angle of 83.8°. Figure 16b illustrates these individual contributions and the corresponding predicted overall ^31^P{^11^B} REAPDOR curves, showing good agreement with the experimental data in both systems. In the case of **1**, the agreement can be improved even further if a *J*-anisotropy of 100 Hz is included in the analysis. Analogous considerations apply for the analysis of ^11^B{^31^P} REDOR data. Three-spin ^11^B^…31^P^…11^B REDOR and REAPDOR curves show a considerable dependence on the angle subtended by the two internuclear vectors [79]. In systems where the details of structural geometry are not known, constraints for such angles can, thus, be developed on the basis of experimental NMR data.

### 3.3. FLP-Cyclotrimer

Treatment of (C_6_F_5_)_2_P–CH(CH_3_) –CH_2_–B(C_6_F_5_)_2_ with 9-borabicyclo[3.3.1]nonane (9-BBN) leads to the exchange of H for one C_6_F_5_ substituent at the borane moiety and subsequent aggregation of this intermediate to the cyclotrimer **3** depicted in Figure 17. The solid-state NMR parameters reveal strong intermolecular P-B covalent interactions, signifying relatively modest FLP reactivity [81]. The ^11^B{^31^P} REDOR and ^31^P{^11^B} REAPDOR data are consistent with the crystallographic distance of 2.006 Å, taking into account a *J*-anisotropy Δ*J* of 100 Hz. The correct value of Δ*J* is found by comparing the experimental data with a theoretical curve calculated on the basis of the crystallographically known internuclear distances and using Δ*J* as an adjustable parameter. Alternatively, Δ*J* can be calculated from first principles. In general, both approaches are found in good agreement [19,20]. The main NMR observable proving the cyclotrimeric structure is the homonuclear ^31^P–^31^P dipole–dipole interaction strength defined by the trigonal geometry with P^…^P distances of 5.758(0) Å. Figure 17b shows that the ^31^P–^31^P DQ-DRENAR curve is quantitatively consistent with this geometry. For a correct reproduction of the experimental DRENAR curve, the ^31^P chemical shift anisotropy (CSA) must be accounted for in the simulation, as it affects the efficiency of dipolar recoupling [73]. In contrast, the intermolecular P^…^P distances between different molecular units in monomeric P/B systems are significantly longer, and no DRENAR effect would be expected.

### 3.4. Dimeric and Trimeric CO Adducts

While the reaction of vicinal B/P FLPs with carbon monoxide results in cooperative binding producing simple cyclic adducts [82], the reaction products of trifunctional P/B/B FLPs offering two such borane Lewis acid centers and one phosphane Lewis base center allow for a rearrangement, leading to the integration of the CO moiety into the FLP backbone. Such compounds show a pronounced tendency to self-assemble into dimeric and trimeric macrocycles (see Figure 18), which depends on the nature of the substituents at the phosphane unit [83].

The structural features of these supramolecular adducts were recently characterized by solid-state NMR [83]. Figure 19a shows the ^11^B MAS NMR spectra of both compounds. The resonance assignments to the two distinct boron species are easily accomplished with the help of DFT calculations for both the ^11^B quadrupolar coupling and the magnetic shielding tensors. Compared to the FLP-like species (B1 and B3), the O-bonded boron nuclei (B2 and B4) are distinguished by significantly stronger EFGs and more positive chemical shift values. In the ^11^B MAS NMR spectra, additional multiplicities are observed due to slight crystallographic non-equivalences of the boron sites. For example, the ^11^B MAS NMR spectrum of the trimeric compound shows two groups of resonances, both containing contributions from three different monomeric units (B2, B4, B6 for the O-bonded boron species and B1, B3, B5 for the FLP-like species) in equal proportions. In the ^31^P MAS NMR spectra (Figure 19b), the two crystallographically different phosphorus positions of the dimeric compound are barely resolved. Again, the spectra show evidence of ^31^P–^11^B indirect spin–spin coupling, revealing a multiplet structure. Thus, the simulation must include separate contributions from the ^10^B and ^11^B isotopologues. This spectrum looks even more complex for the trimeric compound, consisting of the three contributions (P1–P3) for each of the two possible isotopologues of the macrocycle and of a minor impurity. To simplify the complex overall lineshapes, the ^1^H→^31^P{^11^B} CP/MAS NMR spectra (Figure 19c) were also acquired for both samples using swept-frequency two-pulse phase modulation SWFTPPM decoupling of ^1^H and ^11^B nuclei simultaneously [84]. This proved to be useful for developing additional constraints for the line-shape fitting procedure. The rather large indirect spin–spin coupling constants ^1^*J*_iso_(^11^B–^31^P) measured for these macrocycles (70 Hz and 80 Hz for the dimer and the trimer, respectively) demonstrate that the dative B---P bonds are comparatively strong compared to other systems. This observation is consistent with the comparatively small ^11^B nuclear electric quadrupolar coupling constants and rather negative ^11^B chemical shift values observed here in relation to other B/P FLPs. The 2D ^11^B{^31^P} J-resolved MAS NMR experiments depicted in Figure 20 confirm the ^11^B resonance assignments and coupling constants.

Figure 21 shows the 2D heteronuclear correlation spectra for both compounds obtained using the ^11^B{^31^P} CP/refocused INEPT MAS NMR experiment. For the dimeric macrocycle, both cross peaks between P1 and B1 and between P2 and B3 can be identified in Figure 21, while the spectrum of the trimeric compounds shows the expected three cross peaks (P1–B1, P2–B3, and P3–B5) with a good resolution. Additionally, a fourth cross peak between P4 and B7 corresponding to the impurity is clearly evident. Note in particular the improved resolution in the ^31^P dimension compared to the standard ^31^P MAS NMR spectra of Figure 19b. Furthermore, the 2D aspect of this INEPT experiment allows improved resolution in the ^11^B dimension as well. Since the cross peaks are well resolved, ^11^B MAS spectra of the single species can be obtained by analyzing the Fourier transforms and their separated rows along the direct dimension. Figure 22 shows the ^11^B{^31^P} REDOR results of **5**, illustrating a dramatic difference between the B/P FLP-like borane unit B1, B3, B5, and B7 (closest P^…^B distance 2.10 Å) and the O-bonded boron species (closest P^…^B distance 4.38 Å). Figure 23 shows the ^31^P{^11^B} REAPDOR data along with their simulations. As in the previous examples, the REAPDOR response must be considered as the weighted sum of the four possible isotopologues. While an approximate analysis based on the closest distance of 2.10 Å already comes reasonably close, we note that the agreement with the experimental data is significantly improved when including the second-nearest neighbor (4.38 Å) in the analysis.

### 3.5. Tetrameric B/P FLP–CO_2_ Macrocycles.

The six-membered cyclic B/P FLP **6** shown in Figure 24 adds carbon dioxide under mild conditions. Upon crystallization, a unique macrocyclic tetramer **7** with bridging CO_2_ molecules between the individual cyclo-FLP units was obtained [85]. The spectroscopic features of this macrocycle were compared with those of a closely related monomeric B/B/P FLP adduct **8**, which can be stabilized upon addition of one extra equivalent of B(C_6_F_5_)_3_ (BCF) to the starting material (cf. Figure 24).

Figure 25a,b show the ^11^B MAS NMR spectra of the monomeric B/B/P FLP and the tetrameric B/P FLP-CO_2_ macrocycle, respectively. The two boron species detected for compound **8** were assigned to boron in the FLP unit (B1, red curve) and in the BCF ligand (B2, blue curve) according to DFT calculations based on the single crystal structure. The ^11^B nucleus associated with B1 shows an isotropic J-coupling to ^31^P with ^1^J = 42 Hz, as determined by 2D heteronuclear J-resolved MAS NMR spectroscopy (see Figure 26) and confirmed by DFT calculation. There is no J-coupling detected for B2. For compound **7**, the two detected boron species originate from two slightly crystallographically different boron positions in the tetrameric macrocycle. No isotropic J-coupling is observed (and expected by DFT calculation) in this case (data not shown).

For both compounds, the ^31^P MAS NMR spectra show only a single species, even though, in the case of **7**, two crystallographically distinct sites are expected. As illustrated by Figure 27a, the J-coupling with the ^11^B nuclei (I = 3/2) present in **8** manifests itself in a resolved multiplet. Figure 27c,d also reveal a significant difference between the ^31^P chemical shift anisotropy parameters for the two compounds, which was confirmed by DFT calculation, further highlighting the effect of intermolecular association.

The molecular structures of these compounds are further confirmed by ^11^B{^31^P} REDOR experiments (Figure 28). The experimental data are found to be in good agreement with simulations based on the molecular conformation obtained from the single-crystal structures. In the case of **8**, the significant differences in the B^….^P internuclear distances for B1 and B2 are easily detected, while, in the case of **7**, a deconvoluted REDOR analysis is precluded by the close signal overlap. Thus, the observed REDOR behavior corresponds to the average of both individual REDOR curves. In this case, it was again found necessary to conduct a three-spin analysis, including the closest intermolecular B^….^P distances to the next B/P FLP molecule inside the macrocycle.

Figure 29 shows complementary ^31^P{^11^B} REAPDOR data. As each of the ^31^P nuclei interacts with two closest ^11^B spins, the existence of different isotopologues must be considered, as described above. For **8**, these contributions consist of a dominant three-spin component (64.2%, 2.87 and 4.32 Å, ^11^B^…31^P^…11^B angle of 78°, blue curve) and two minor two-spin components from the two possible ^11^B–^31^P–^10^B isotopologues (15.9% each), while 4% of the molecules do not contain any ^11^B and, thus, will yield no REAPDOR effect. Analogous simulations were made for **7**, based on the intramolecular (3.29 Å) and the intermolecular closest ^31^P^…11^B distances (4.16 Å), as well as a ^11^B^…31^P^…11^B angle of 119°. Figure 29 shows overall good agreement with the experimental data.

The aggregated character of **7** can be verified by the ^31^P–^31^P CT-DRENAR data shown in Figure 30. The monomeric FLP and the macrocyclic FLP tetramer are well differentiated based on the strength of the homonuclear ^31^P–^31^P magnetic dipole–dipole interactions. For compound **8**, the closest internuclear ^31^P^….31^P distance is 8.77 Å and, thus, the dipolar recoupling effect (the difference signal intensity *ΔS*) is very small, in very good agreement with the simulation (Figure 30a, green curve). In contrast, each ^31^P nucleus in the tetramer **7** is neighbored by two ^31^P nuclei at the distance of 6.16 Å, leading to a significantly stronger recoupling effect, which is again in excellent agreement with the simulation (Figure 30b, green curve). Moreover, the experimental data also differentiate this material very well from a hypothetical FLP dimer, for which only a two-spin interaction would have to be considered.

### 3.6. An FLP-Cyclooctamer

The six-membered cyclic B/P FLP **6** illustrated in Figure 31 shows a remarkable tendency for self-assembly at low temperatures in solution and upon crystallization, producing the cyclooctameric structure **9** with alternating chair- and boat-like conformations [86]. The boron and the phosphorus atoms are placed at the tip of the chair (B1 and P1), while, in the boat conformation, the FLP centers are part of the basis (B2 and P2) (see Figure 32). To facilitate the computation of the NMR observables, the simplified molecules **10a** and **10b** served as models of the corresponding cutouts of the molecular structure of **9**. The ^11^B MAS and the ^31^P{^1^H} CP/MAS NMR spectra are shown in Figure 31c,d. The ^11^B MAS NMR spectrum features two signals at isotropic chemical shifts of −3.7 ppm and −5.4 ppm. As already shown in Figure 6, the resolution can be significantly improved by the triple-quantum single-quantum correlation experiment; the INEPT method offers an alternative (*vide infra*). Each of the two resonances in the ^31^P{^1^H} CP/MAS NMR spectra centered at 20.0 ppm and 17.7 ppm in Figure 31d consists of two distinct isotopologue signals (due to the effect of ^31^P–^11^B and ^31^P–^10^B J-coupling). The four crystallographically distinct FLP monomers of **9** in the same conformation (chair- or boat-like) present common NMR signals [86].

The assignment of experimental to calculated ^11^B NMR parameters and, therefore, to certain conformers of the monomeric unit is not straightforward, since the calculated ^11^B chemical shifts and quadrupolar coupling parameters for the two boron sites are very similar. Key to the assignment is the considerable difference in the ^31^P chemical shift anisotropies between the P1 and P2 sites evident in Figure 33, which is also consistent with the DFT calculations. This difference is quite evident in the ^31^P MAS NMR spinning sideband pattern, which extends over a significantly wider spectral range for the P1 (chair) than for the P2 (boat) unit. As each ^11^B resonance could be assigned to the directly bound phosphorus atom using the 2D ^31^P{^11^B} CP/refocused INEPT correlation experiment (see Figure 34), the boron species at −3.8 ppm can be attributed to boron in a chair conformation (B1) of the monomeric unit, while the boron nucleus in the boat conformation (B2) gives rise to a resonance at about −5.3 ppm. The latter species experiences a slightly stronger quadrupolar coupling, as also confirmed by DFT calculations. Obviously, if the crystal structure of the cyclooctamer were unknown, the experimental data discussed above would also allow for alternative interpretations. For example, the two pairs of ^11^B and ^31^P signals could equally well be a result of a dimeric or tetrameric macrocycle, or a result of two different crystal structures of the FLP monomer, which are mixed inside the powder sample. To exclude the latter scenario, a ^31^P CP/BaBa spectrum was acquired, in which cross peaks arise for directly dipolar coupled nuclei. With the objective of selecting correlation peaks only for the most proximal ^31^P nuclei, this experiment was done with the minimum number of excitation blocks, using the BaBa-XY16 sequence. Under this condition, only the P1–P2 cross correlation (*r*(P1–P2) = 5.3 Å) is observed (see Figure 35). In contrast, the dipolar couplings between the chemically equivalent ^31^P species are much weaker (*r*(P1–P1) = 8.4 Å; *r*(P2–P2) = 10.1 Å and, thus, no auto-correlation peaks are seen. If the pairs of boron and phosphorus resonances would result from two different crystalline polymorphs, no cross-correlations would be observed.

Figure 36a shows the ^11^B{^31^P} REDOR curve of **9**. Since both ^11^B species are strongly superimposed in the ^11^B MAS NMR spectrum, only the total signal intensity can be analyzed. Alongside the experimental data, three different simulated REDOR curves are depicted, which are based on the single-crystal structure obtained via X-ray diffraction. It is expected that the dephasing of the ^11^B resonances is dominated by dipolar couplings to the covalently bound ^31^P atom of the neighboring FLP monomer (green curve, average internuclear B^…^P distance of 2.13 Å (B^…^P distances are 2.15 Å (B1) and 2.11 Å (B2)). If no oligomer were formed, the strongest dipolar coupling would result from the intramolecular B–P interaction, for which another simulation based on an average distance of 3.42 Å is depicted (blue curve). A comparison with the experimental data clearly proves the formation of a supramolecular structure, where strong intermolecular ^11^B/^31^P dipole–dipole interactions are present. Regarding this comparison, an excellent agreement between simulated and experimental data is only achieved if a full three-spin system is included in the simulation (black curve). The angle between both dipolar vectors used for this simulation was obtained from the crystal structure (B1: 145° and B2: 144°). The excellent agreement between experimental data and simulation proves that considering the interaction with the two closest spins is sufficient to quantitatively reproduce the REDOR data in this case. The analogous ^31^P{^11^B} REAPDOR data, shown in Figure 36b, produce the same conclusion. As discussed above, the experimental REAPDOR curve is the sum of the three distinct contributions arising from the boron isotopologues in the expected 64.2:15.9:15.9 ratio.

Finally, the ^31^P–^31^P CT-DRENAR experiment can be used to confirm the macrocyclic structure of **9**. In analogy to the tetrameric compound **7**, only three-spin simulations based on the actual crystal structure of **9** are suitable to represent the experimental data, directly excluding mono- and dimeric structures. Interestingly, P1 shows a much stronger dephasing than P2 despite the identical internuclear ^31^P^…^^31^P distances of 5.3 Å. This can be explained by the strongly differing ^31^P chemical shift anisotropies evident in Figure 33, which are known to influence DRENAR experiments [72,73,74,75]. For the octameric structures, it is important to determine if this influence originates from the ^31^P chemical shift anisotropy (CSA) of the coupling partners or the observed nucleus itself. As can be deduced from the simulations in Figure 37c, an increased ^31^P CSA of the coupling partners strongly decreases the maximum DRENAR difference signal, while corresponding changes in the ^31^P CSA of the observed nucleus show no influence within the regime of experimentally determined Δ*σ* up to 124 ppm. Therefore, P2 interacting with two P1 phosphorus spins with a large ^31^P CSA of 124 ppm shows a weaker DRENAR effect compared to P1 interacting with two P2 phosphorus spins (CSA = 78 ppm). To increase the overall DRENAR effect, longer dipolar mixing times can be applied for weakly interacting ^31^P nuclei, as demonstrated in Figure 37b. Again, a good agreement between three-spin simulations and experimental data confirms the findings discussed. In conclusion, the CT-DRENAR experiments are well suited to determine ^31^P^…^^31^P internuclear distances and spin system geometries, as long as CSA effects are taken into account in the simulation. This requires that the latter can be determined accurately from slow-spinning MAS NMR spectra.

## 4. Conclusions

Advanced solid-state NMR experiments can provide key information regarding the structure and bonding character of aggregated FLPs and FLP adducts. If aggregation proceeds via intermolecular association of the borane/phosphane units, ^11^B chemical shift and nuclear electric quadrupolar coupling constants, as well as isotropic ^1^*J* spin–spin coupling constants, are particularly useful in characterizing the strength of such intermolecular associations. Table 1 gives a comprehensive summary of all the relevant NMR observables obtained in this field.

Likewise, REDOR and REAPDOR offer the possibility of distance measurements. For the analysis of such experiments on FLP (adduct) aggregates, two-spin models are usually no longer appropriate, and the full distance geometry of the heteronuclear spin system must generally be taken into account. In many cases, the approximate analysis in terms of three-spin systems involving the shortest distances tends to be an acceptable approximation. Detailed systematic studies on model compounds indicate that the sensitivity of REDOR and REAPDOR methods is limited to maximum distances of about 6 Å; measurements toward longer distances in FLP-related compounds are impeded by spin–spin relaxation effects [79]. As shown in the example of the tetrameric CO_2_ adduct **7** and the octameric FLP **9**, the evaluation of homonuclear ^31^P–^31^P dipole–dipole interaction strengths via DQ-DRENAR, and other homonuclear recoupling experiments are particularly useful for differentiating between different models of intermolecular associations. While, in the present study, all the compounds discussed could be characterized independently by single-crystal X-ray diffraction, the NMR methodologies and database developed here will be of great value for learning about the structural organization of poly- and nano-crystalline materials or surface FLP deposits to be studied in future endeavors, including, for instance, the development of heterogeneous FLP catalysts. Initial efforts in this direction are already appearing in the literature. For example, a heterogeneous H_2_ splitting catalyst based on an intermolecular FLP was designed by reacting an organoboron Lewis acid covalently attached to a silica support with the Lewis base P(*t*-Bu)_3_ [87] and characterized by standard MAS NMR methods. Introducing more advanced NMR methodologies to such systems will certainly be of great benefit in characterizing their more intricate structural features. Ultimately, the structure/function correlations discovered along these lines will be useful for optimizing their catalytic performance.

## 5. Materials and Methods

Solid-state MAS NMR studies were conducted on a Bruker Avance III 300 spectrometer, and DSX 400 and 500 MHz spectrometers (magnetic flux density of 7.05, 9.39, and 11.74 T, respectively). Typical MAS frequencies were 12.5 kHz in 4-mm NMR double- and triple-resonance probes. ^1^H decoupling was typically used applying the SWFTPPM decoupling scheme [84] with 60 kHz nutation frequency. ^31^P{^1^H} CP/MAS NMR spectra were typically acquired via ramped (90%–100% of maximum Hartmann–Hahn nutation frequency of 43 kHz) cross-polarization [88] at contact times of 500 µs and typical relaxation delays of 15 s. All the two-dimensional J-resolved, correlation, and dipolar recoupling experiments were measured at MAS frequencies of 10–15 kHz and nutation frequencies near 50 kHz. Stationary magnetization conditions were ensured by saturation combs preceding the pulse sequences. The first and second J-evolution periods in the INEPT block were optimized for maximum signal intensity giving delays of 4.3 and 5.0 ms, far below the value suggested by 1/(2J) to avoid signal loss due to transverse relaxation. Spectral deconvolution was done with the DMFIT software [39] (version 2011).

DFT calculations of NMR parameters were conducted based on crystallographic input or geometry-optimized molecular models such as those shown in Figure 30 (TURBOMOLE (version 6.5 and 7.1) [43,89] combined with the TPSS functional [90] and an Ahlrich’s def2-TZVP basis set [91]. Chemical shift calculations also used TURBOMOLE and the def2-TZVP basis set in combination with the B3LYP functional [92,93]. Quadrupolar coupling parameters were calculated on a GGA DFT level using GAUSSIAN (version GAUSSIAN09) [42] and the B97-D functional [94]. The def2-TZVP basis set obtained from the EMSL database [95,96] could be used with additional functions for boron from the cc-pCVTZ basis set to enhance the accuracy near the nucleus. Anisotropic J-coupling constants were calculated with the software CPL implemented in the ADF program [97,98,99]. The calculation was performed using the PBE0 hybrid functional [100] and JCPL basis sets [101], which are based on the TZ2P ZORA basis set, but which also include additional functions with high exponents for an improved representation of the electron density close to the nuclei.

## Figures and Tables

**Figure 1 molecules-25-01400-f001:**
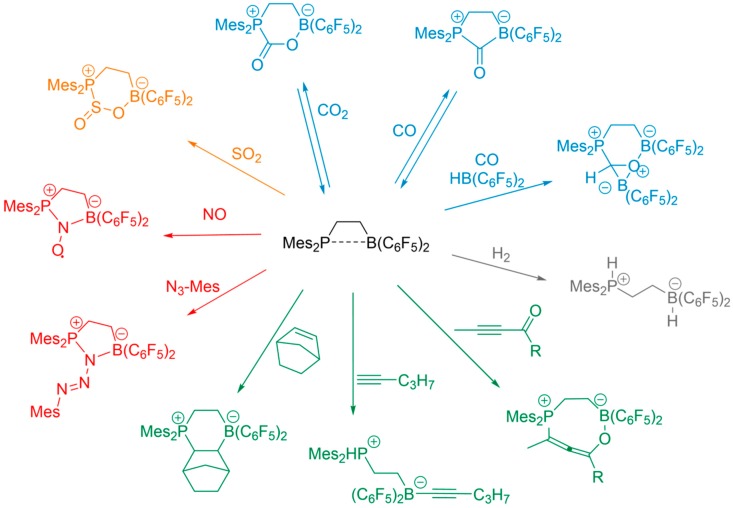
Schematic overview for various chemical reactions performed with intramolecular borane–phosphane (B/P) frustrated Lewis pairs (FLPs).

**Figure 2 molecules-25-01400-f002:**
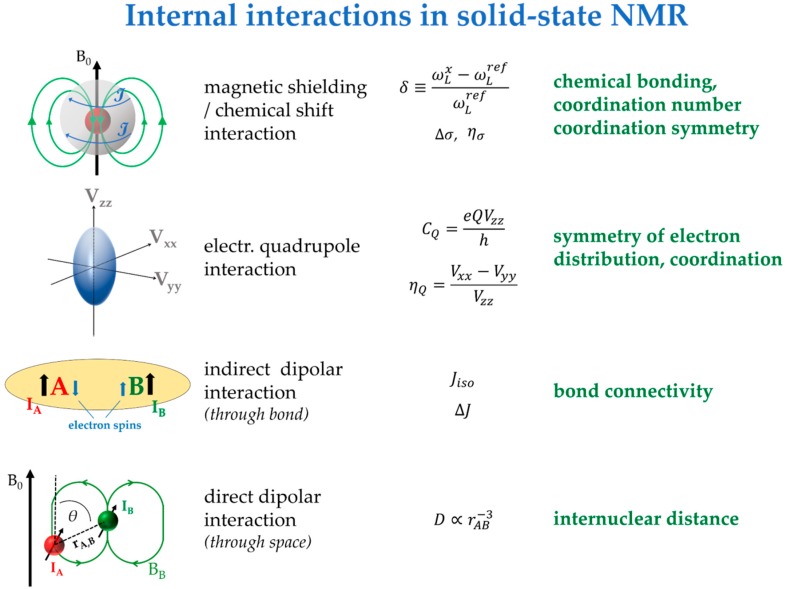
Anisotropic interactions influencing the appearance of solid-state NMR spectra for diamagnetic solids, parameters extractable from the NMR spectra describing these interactions, and the structural information they are connected to.

**Figure 3 molecules-25-01400-f003:**
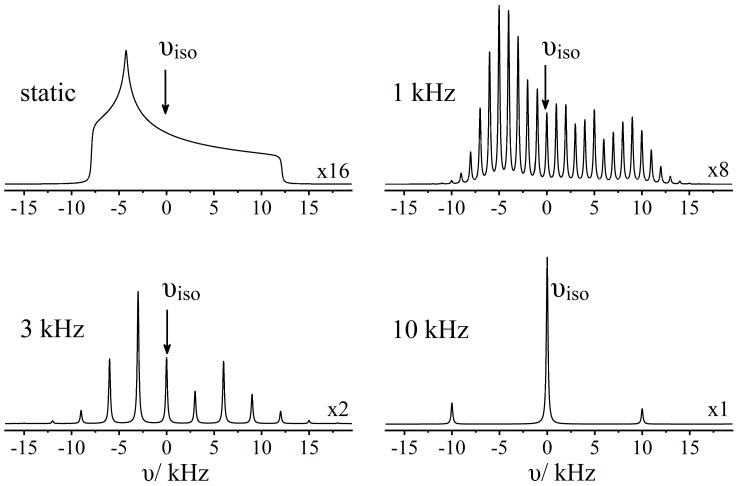
Numerical simulations demonstrating the effect of magic-angle spinning (MAS) upon the solid-state NMR line shape of a spin-1/2 nucleus with dominant line broadening due to the magnetic shielding anisotropy. Spinning sidebands are located at integer multiples of the rotor frequency.

**Figure 4 molecules-25-01400-f004:**
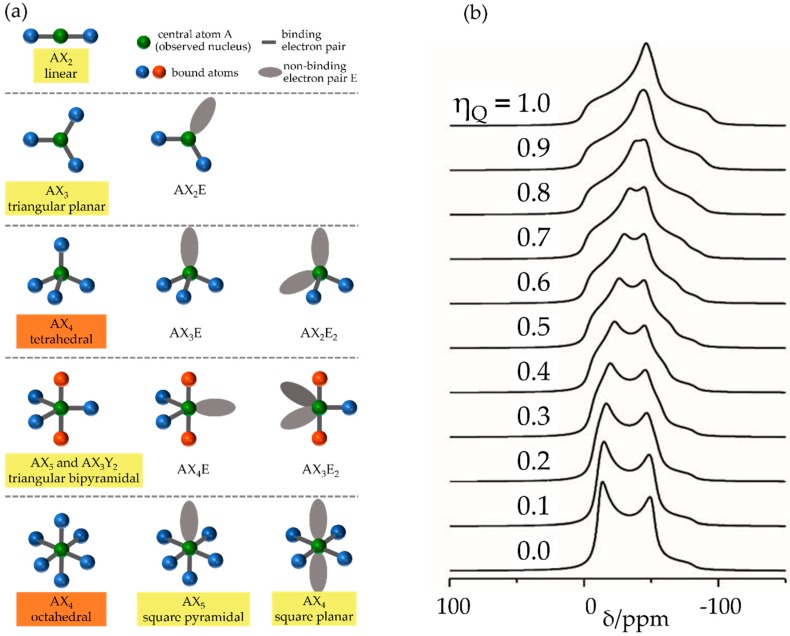
(**a**) Typical bonding geometries realized in molecular inorganic chemistry and characterization of their local symmetries. Geometries underlined in orange produce isotropic magnetic shielding tensors (Δ*σ* = 0) and zero electric field gradients ($V_{zz}=V_{xx}=V_{yy}=\eta_{\sigma }=$ 0; hence, $C_{Q}=\eta_{Q}=0$) at the central nucleus under consideration. Geometries underlined in yellow produce axially symmetric magnetic shielding and EFG tensors ($\eta_{\sigma }$ and $\eta_{Q}\;$= 0). Geometries not underlined produce asymmetric magnetic shielding and electric field gradient tensors (0 ≤ ($\eta_{\sigma },\;\eta_{Q}$) ≤ 1). (**b**) Typical MAS NMR line shapes predicted by second-order perturbation theory for the *m* = ½ <−> *m* = −½ central transitions of half-integer spin nuclei due to quadrupolar interaction with EFGs of different $\eta_{Q}$ values.

**Figure 5 molecules-25-01400-f005:**
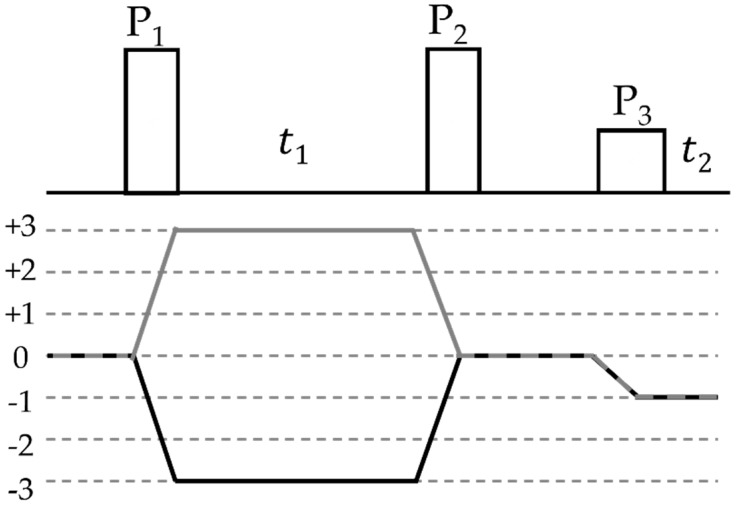
Pulse sequence and coherence path diagram selected via the phase cycling scheme of a triple-quantum MAS-NMR experiment. For spin 3/2 nuclei, coherence orders ranging from +3 to −3 are possible.

**Figure 6 molecules-25-01400-f006:**
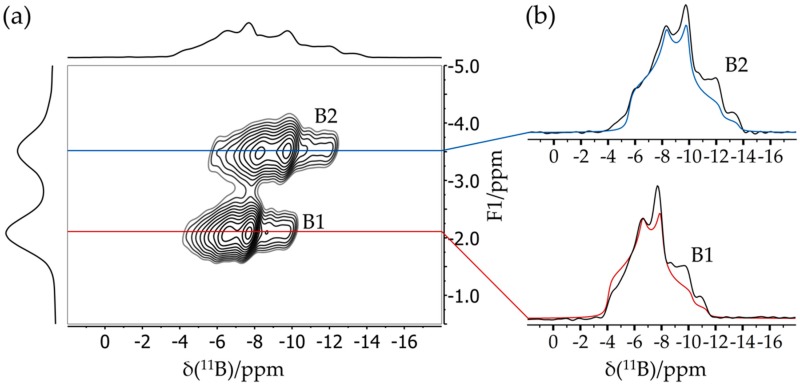
**(a)** Typical TQ MAS spectrum of an FLP octamer featuring two distinct boron sites B1 and B2. While the spectrum in the horizontal dimension is equivalent to the regular MAS spectrum, the spectrum in the vertical (F1) dimension is purely isotropic, yielding well-separated symmetric line shapes. (**b**) Corresponding F2 sections representing resolved MAS spectra of the individual boron sites including line shape simulations (see Section 3.6).

**Figure 7 molecules-25-01400-f007:**
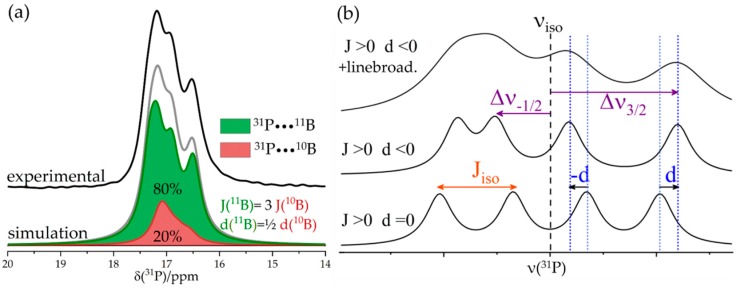
(**a**) Exemplary ^31^P MAS NMR spectrum of a B/P FLP with isotropic ^31^P J-couplings to ^11^B (green) and ^10^B (red) of 33 and 11 Hz, respectively, and residual dipolar couplings of 8 and 16 Hz. (**b**) Simulated MAS NMR spectra demonstrating the general shape for an *S* = ½ nucleus coupled to a quadrupolar *I* = 3/2 nucleus. Bottom spectrum: no quadrupolar interaction. Middle spectrum: with quadrupolar interaction, leading to an additional dipolar splitting parameter *d* arising from cross-terms between the quadrupolar interaction and the heteronuclear dipolar interaction. The dashed blue lines on the right hand side, each located at a peak maximum of the bottom or middle curve, show the relative shift of two resonances by the residual dipolar coupling *d*, which is negative for coupling of S to I with *m_I_* = ±1/2 and, vice versa, positive for *m_I_* = ±3/2. The overall change of the resonance frequency *Δν*_m_ caused by isotropic *J*-coupling and residual dipolar coupling *d* is exemplarily shown for resonances originating from couplings of S to I with *m_I_* = −1/2 and *m_I_* = +3/2. Top spectrum: simulated profile with additional line broadening to create a line shape similar to the experimental curve (shown in green in (**a**)).

**Figure 8 molecules-25-01400-f008:**
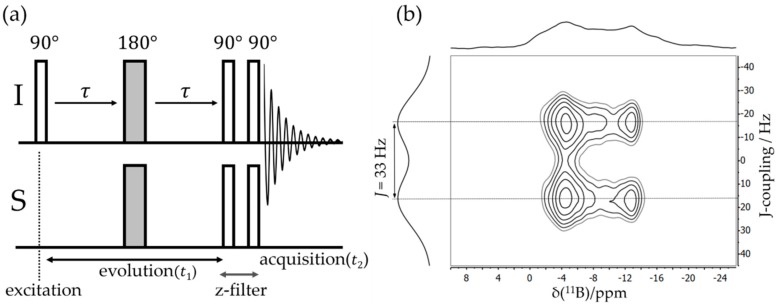
(**a**) Schematic representation of the pulse sequence for heteronuclear-J-resolved spectroscopy. (**b**) A typical two-dimensional (2D) ^11^B{^31^P} J-resolved MAS NMR spectrum of a B/P FLP. In the direct dimension, the horizontal projection is equal to the ^11^B MAS NMR spectrum, while the spectrum obtained from Fourier transforming the data in the indirect dimension shows the doublet splitting due to J-coupling. The dipolar evolution time stated in the text is *t*_1/2_ = *τ* + ½ × *t*_180°_ where *t*_180°_ is the 180° pulse length.

**Figure 9 molecules-25-01400-f009:**
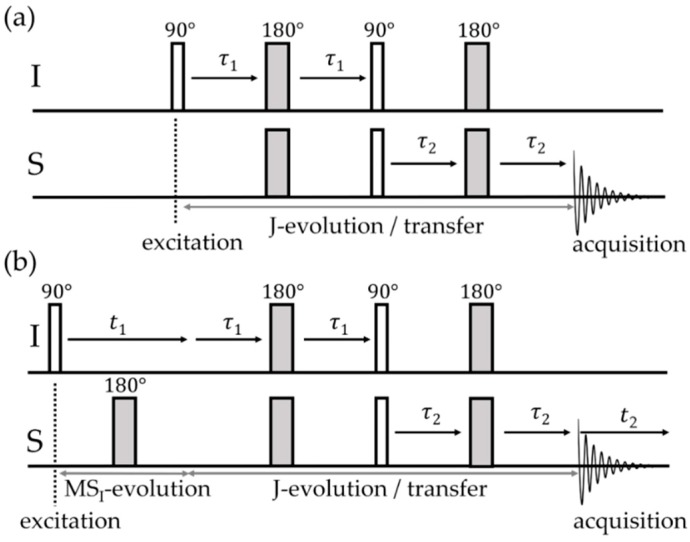
One-dimensional (1D) (**a**) and 2D (**b**) refocused insensitive nuclei enhanced by polarization transfer (INEPT) experiment used for polarization transfer developed via J-coupling to detect heteronuclear correlations.

**Figure 10 molecules-25-01400-f010:**
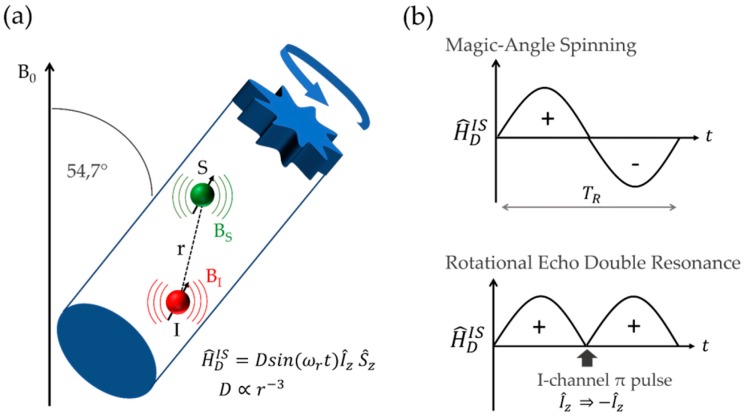
Principle of the rotational echo double resonance (REDOR) experiment. (**a**) Schematic representation of the dipolar interaction of two spins I and S inside a spinning rotor inclined at the magic angle. Local magnetic fields produced by their magnetic moments (*B*_I_ and *B*_S_) modify the precession frequency of the interacting nuclei. (**b**) Under MAS conditions, the heteronuclear dipolar Hamiltonian ${\hat{H}}_{D}^{IS}$ oscillates sinusoidally with rotor orientation, leading to cancellation upon completion of the rotor cycle in the fast spinning limit. Sign inversion created by a π-pulse applied to the non-observed I spins (turning *Î_z_* into *−Î_z_*) interferes with this cancellation (bottom), resulting in the dipolar re-coupling effect.

**Figure 11 molecules-25-01400-f011:**
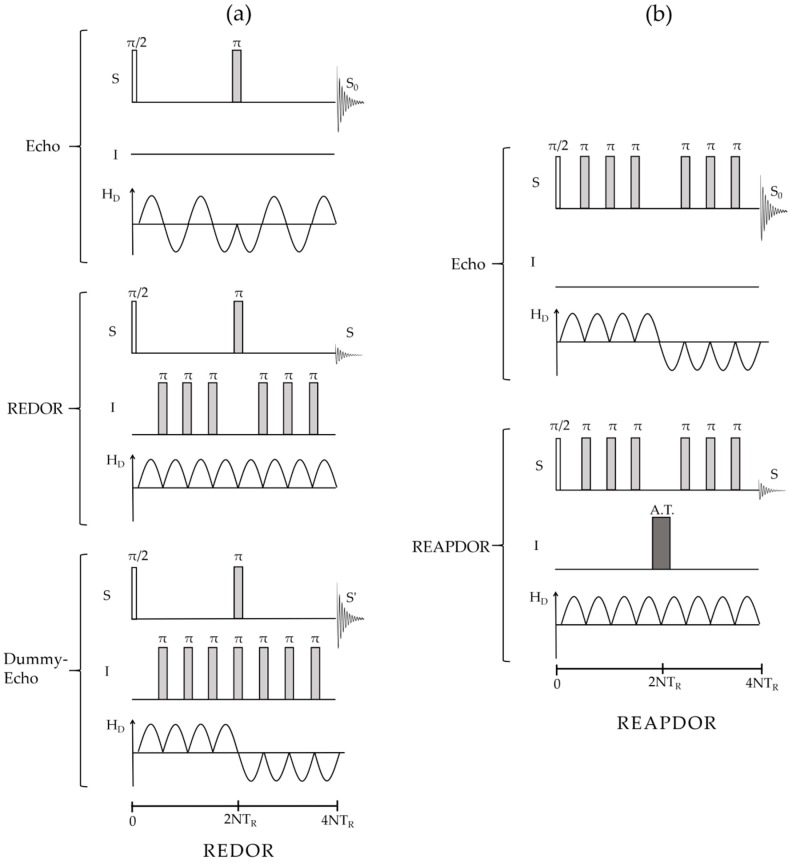
(**a**) Pulse sequence elements of the standard compensated REDOR experiment. The evolution of the dipolar Hamiltonian is shown below each pulse sequence element. Top: spin-echo experiment on the observed S-channel to acquire the reference intensity *S*_0_. Middle: echo experiment with dipolar recoupling, resulting in the intensity *S*_._ Bottom: “dummy” echo experiment producing a correction *S*’ to the dephased intensity to account for pulse inaccuracies and finite pulse length effects. (**b**) Pulse sequence for the rotational echo adiabatic passage double resonance (REAPDOR) experiment used for recoupling of the direct dipolar interaction to quadrupolar nuclei. Here, an adiabatic transfer (AT) pulse inverts magnetization on the I-channel.

**Figure 12 molecules-25-01400-f012:**
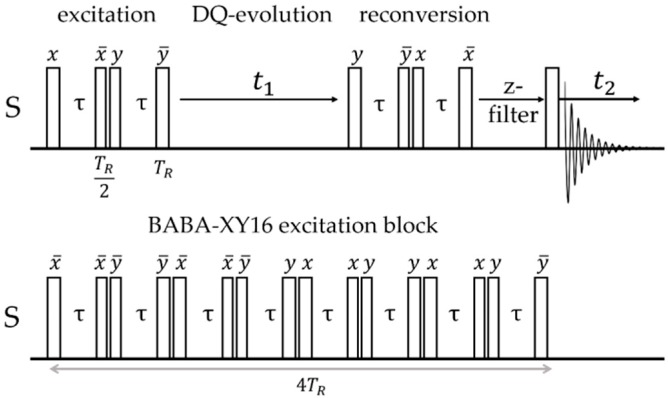
Top: 2D double quantum/single quantum NMR spectroscopy with *back-to-back* (*BaBa*) excitation. All pulses have a 90° flip angle. Bottom: BaBa-XY16 excitation for increased efficiency and excitation bandwidth. The symbols *x* and *y* denote pulse phases.

**Figure 13 molecules-25-01400-f013:**
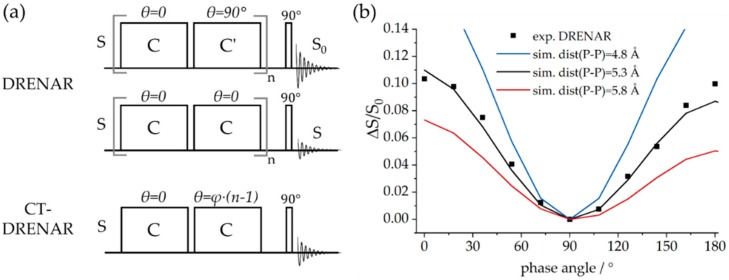
(**a**) Dipolar re-coupling effects nuclear alignment reduction (DRENAR) (top) and constant-time (CT)-DRENAR (bottom) experiments based on reduction of longitudinal magnetization by excitation of double quantum (DQ) coherence using the homonuclear dipolar coupling interaction. While, in the standard DRENAR experiment, the indirect dimension is incremented by adding DQ excitation blocks, in CT-DRENAR, the relative phase of the second excitation block is incremented by an arbitrary angle φ. (**b**) Exemplary CT-DRENAR spectrum for one phosphorus species in an FLP macrocycle further discussed in Section 3.6. Additionally, simulated CT-DRENAR curves for internuclear distances of 4.8, 5.3, and 5.8 Å are depicted.

**Figure 14 molecules-25-01400-f014:**
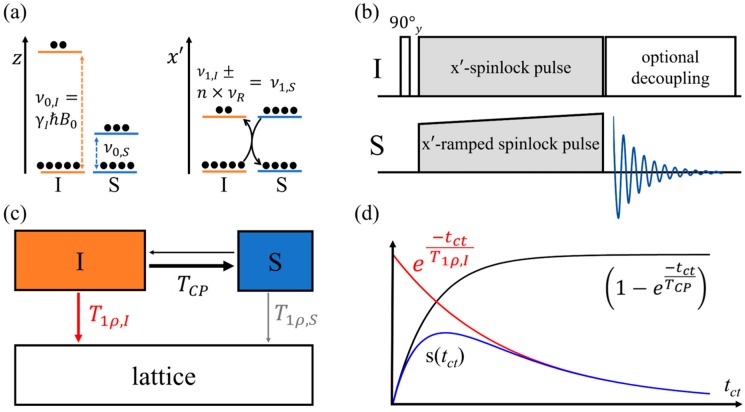
Principle of the cross-polarization (CP) NMR experiment: (**a**) Zeeman states for an I (orange) and an S nucleus (blue) along the quantization axis, split by the external magnetic field (left) and by the magnetic component *B*_1_ of the applied radiofrequency field under spin-lock conditions. Here, full circles illustrate population differences. (**b**) Pulse sequence timing diagram. During the spin-lock time, the nutation frequency of one of the nuclear species is ramped to account for modulation effects in the Hartmann–Hahn matching condition caused by MAS. (**c**) Competitive cross-relaxation and rotating frame spin-lattice relaxation processes. (**d**) Signal intensity as a function of contact time resulting from these competitive processes.

**Figure 15 molecules-25-01400-f015:**
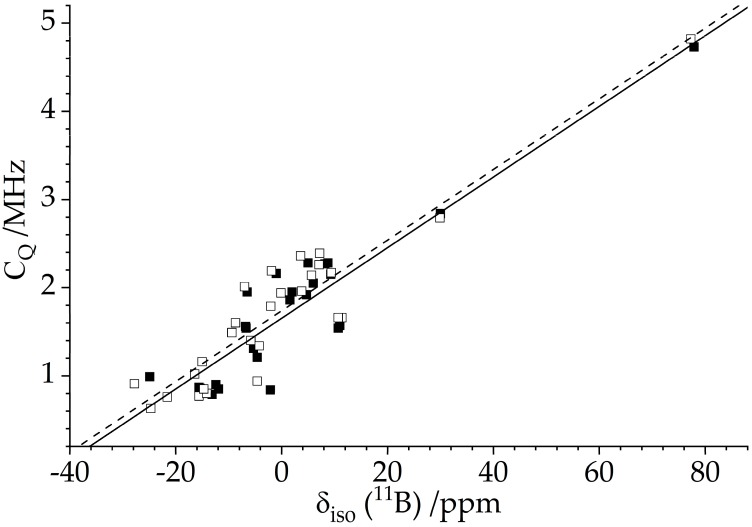
Correlation of ^11^B isotropic chemical shifts and ^11^B nuclear electric quadrupole coupling constants for vicinal B/P FLPs [78,79]. Closed symbols, solid line (*C*_Q_ = 1.652 MHz + 0.040 × *δ*/ppm; *R^2^* = 0.84): experimental data; open symbols, dashed line (*C*_Q_ = 1.738 MHz + 0.040 × *δ*/ppm; *R^2^* = 0.87): DFT-calculated values.

**Figure 16 molecules-25-01400-f016:**
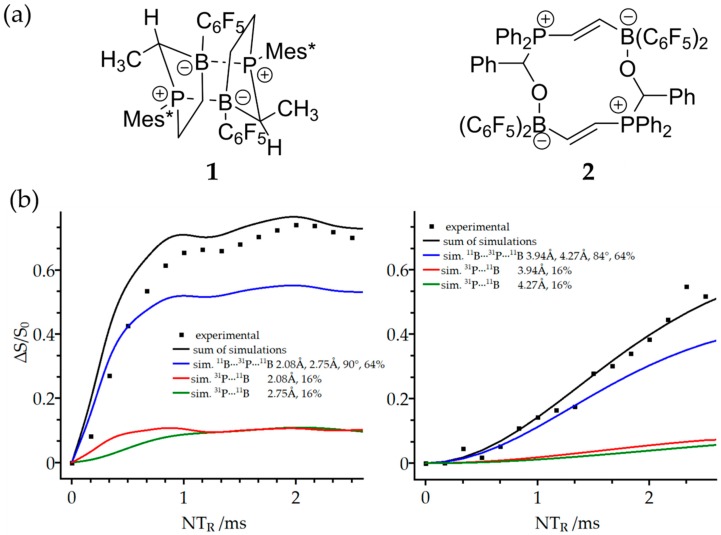
(**a**) Schematic representation of B/P FLP dimer and dimeric adduct of a B/P FLP with benzaldehyde and (**b**) characterization of both compounds using ^31^P{^11^B} REAPDOR experiments. The complete simulation and its three isotopologue constituents are shown as summarized in the insets [79].

**Figure 17 molecules-25-01400-f017:**
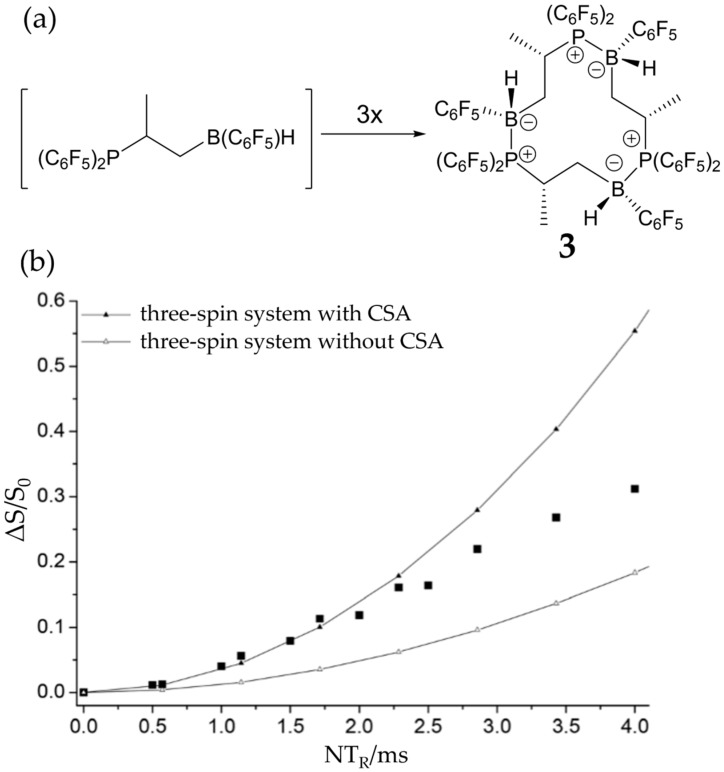
(**a**) Formation of FLP-cyclotrimer **3** and (**b**) ^31^P–^31^P DRENAR build-up curve for **3**. Solid squares: experimental data points. Solid curves: simulated DRENAR curve based on the intermolecular dipole–dipole coupling only (lower curve) and with additional consideration of the ^31^P CSA (Δ*σ* = 99 ppm), leading to an improved agreement with the experimental data. Reproduced from Reference [81].

**Figure 18 molecules-25-01400-f018:**
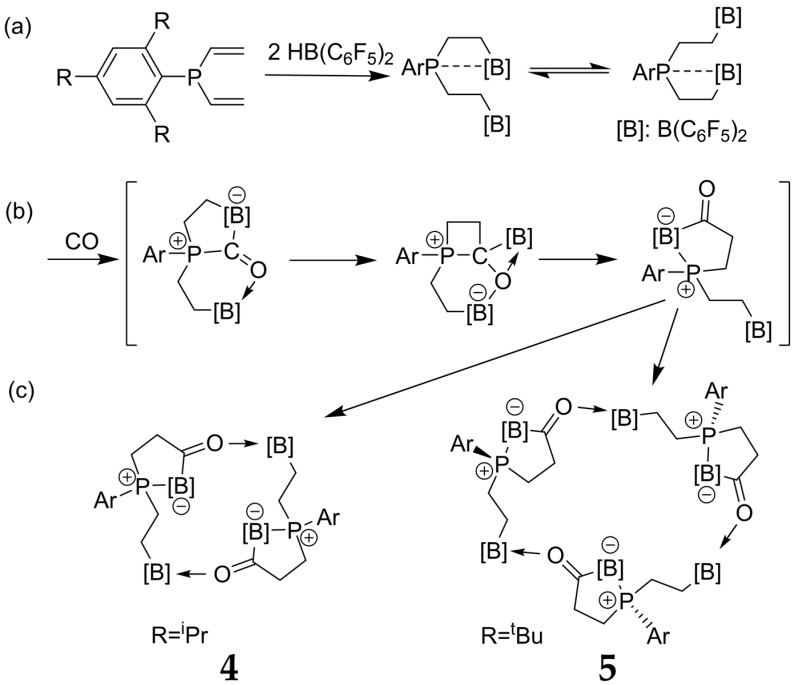
Preparation routes and molecular structures of B/P FLP–CO dimeric and trimeric macrocycles. (**a**) Synthesis of a B/P FLP with two borane Lewis centers; (**b**) reaction of this B/P FLP with CO, resulting in a five-membered intermediate, which (**c**) can self-assemble to dimeric or trimeric adducts.

**Figure 19 molecules-25-01400-f019:**
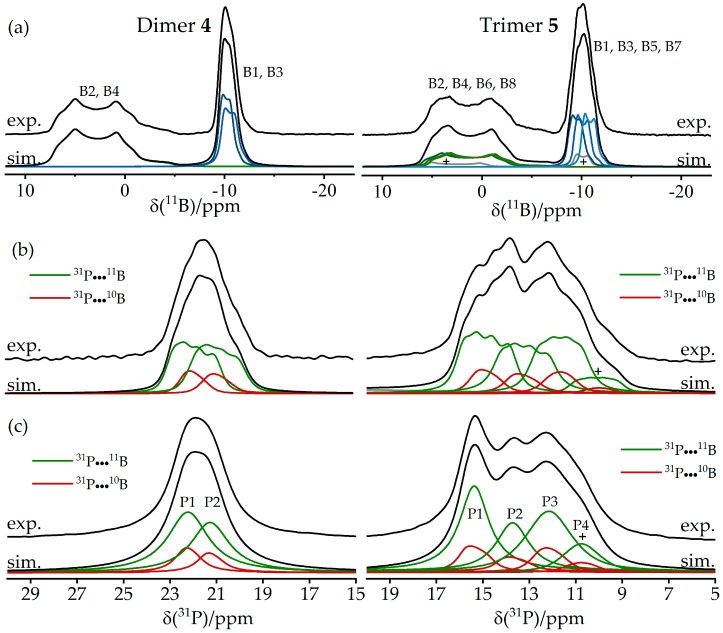
Experimental and simulated (**a**) ^11^B MAS, (**b**) ^31^P MAS, and (**c**) ^1^H→^31^P{^11^B dec} CP/MAS NMR spectra of the B/P FLP–CO dimer **4** and trimer **5**. Simulated ^31^P MAS NMR spectra include the individual contributions for ^11^B (green) and ^10^B (red) isotopologues. Note the effect of ^11^B decoupling suppressing the multiplet structure due to ^31^P–^11^B indirect spin–spin coupling in (c). Impurities are marked with a cross. In (a) the signals labeled B7 and B8 originate from an impurity.

**Figure 20 molecules-25-01400-f020:**
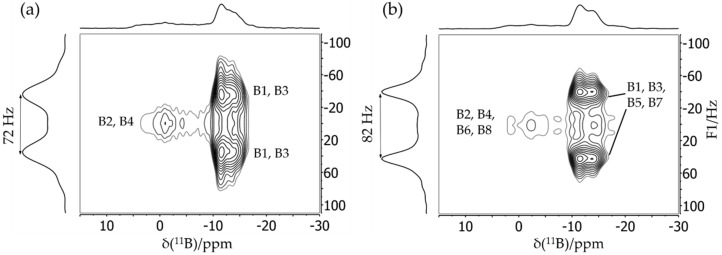
The 2D ^11^B{^31^P} J-resolved MAS NMR spectra of (**a**) **4** and (**b**) **5**. In (**b**), the additional sites B7 and B8 originate from an impurity.

**Figure 21 molecules-25-01400-f021:**
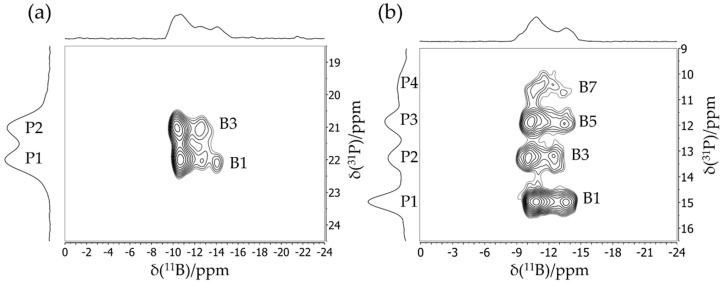
^11^B{^31^P} CP/refocused INEPT MAS NMR spectra of (**a**) **4** and (**b**) **5**. The sites P4 and B7 in (**b**) arise from an impurity.

**Figure 22 molecules-25-01400-f022:**
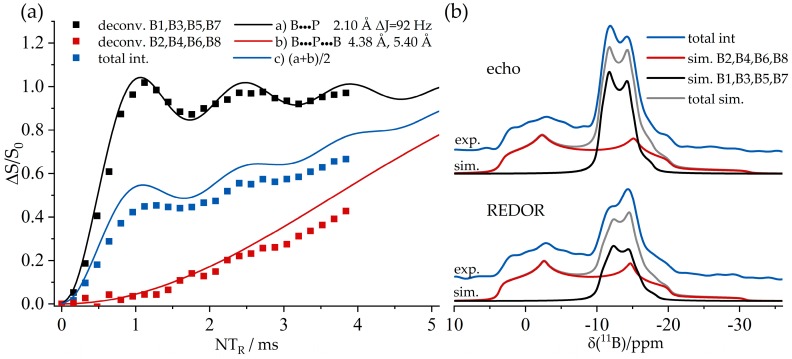
(**a**) Experimental ^11^B{^31^P} REDOR curves (squares) obtained for **5**. Data represented by the black and red squares were obtained by line-shape deconvolution of every 1D REDOR (S) and echo spectrum (S_0_) (see Figure 11), using a single line for each of the two boron environments, i.e., for boron species bound to phosphorus (black) and oxygen (red). Additionally, the intensity of an integral over the total superimposed spectrum is depicted (blue squares). Corresponding simulations based on the single-crystal structure are included as solid lines. The simulations consider a two-spin P–B system with a distance of 2.10 Å (black curve) and a three-spin ^11^B^…31^P^…11^B system with distances of 4.38 and 5.4 Å and a B^…^P^…^B angle of 117° (red curve). Furthermore, the normalized sum of both simulations is shown (blue curve). In (**b**), the REDOR (bottom) and the reference spectra (top) acquired for a dipolar evolution time of 0.48 ms are depicted along with corresponding line-shape simulations.

**Figure 23 molecules-25-01400-f023:**
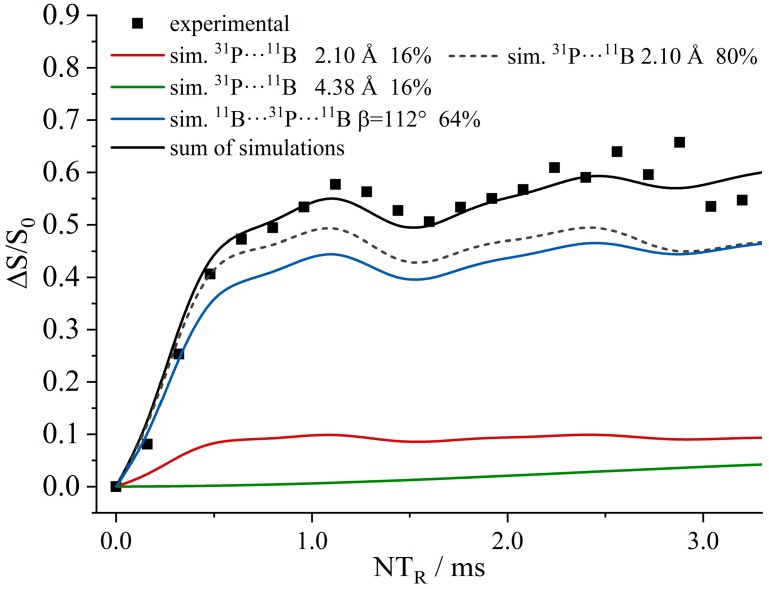
Experimental and simulated ^31^P{^11^B} REAPDOR curves obtained for **5**. Corresponding simulations based on the single-crystal structure are included as solid lines. The simulated curve is the weighted sum of the individual curves obtained for the three possible isotopologues: a three-spin ^11^B^…31^P^…11^B system with distances of 2.10 and 4.38 Å (64.2%, red curve) and a B^…^P^…^B angle of 112° and two ^11^B^…31^P two-spin systems with distances of 2.10 and 4.38 Å (blue and green curves). The dashed curve is an approximate two-spin simulation, neglecting the secondary 4.38 Å distance.

**Figure 24 molecules-25-01400-f024:**
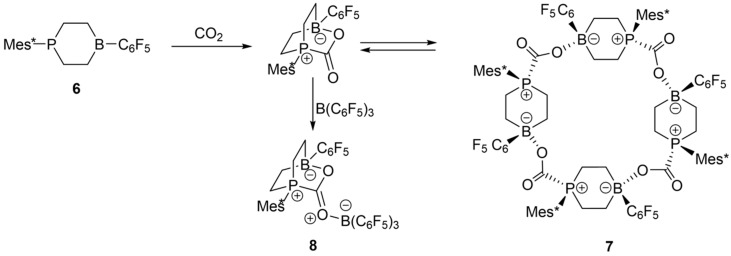
Preparation and molecular structure of the tetrameric B/P FLP–CO_2_ adduct **7** and structure of the monomeric B(C_6_F_5_)_3_ and CO_2_ adduct **8.**

**Figure 25 molecules-25-01400-f025:**
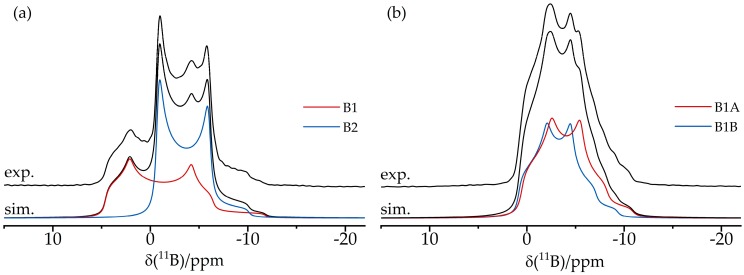
Experimental and simulated ^11^B MAS NMR spectra of (**a**) the monomeric compound **8** and (**b**) the tetrameric FLP–CO_2_ adduct **7**.

**Figure 26 molecules-25-01400-f026:**
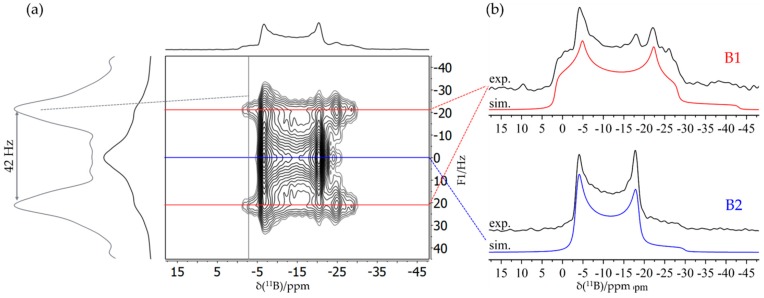
(**a**) The 2D ^11^B{^31^P} heteronuclear J-resolved MAS NMR spectrum of **8**. In addition to the positive projections of the indirect dimension, the column extracted at −2.5 ppm is depicted on the left-hand side. A comparison of these show that the species B1 shows J-coupling with ^31^P nuclei, characterized by an isotropic coupling constant of 42 Hz, while no J-coupling is observed for species B2. Hence, the 2D experiment can be used to resolve both species. This is demonstrated in (**b**), showing the sum of rows extracted at −21 Hz and 21 Hz for species B1 and the row extracted at 0 Hz for B2.

**Figure 27 molecules-25-01400-f027:**
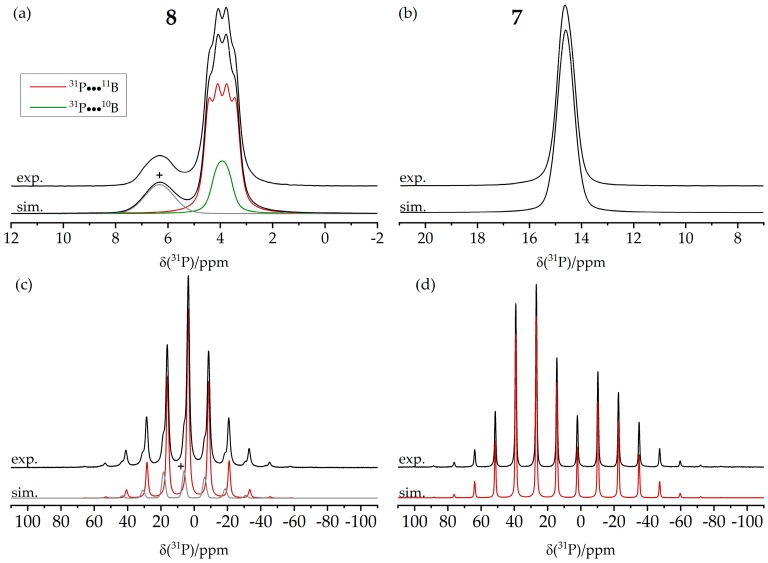
Experimental and simulated ^31^P{^1^H} CP/MAS NMR spectra of **8** and **7** measured at 7.05 T and MAS frequencies of (**a**, **b**) 12.5 kHz and (**c**, **d**) 3.0 kHz. Note the significant differences in the spinning sideband patterns, indicating the potential of the ^31^P chemical shift anisotropy for site differentiation. The minor component observed in **8** is attributed to an impurity (marked by a cross).

**Figure 28 molecules-25-01400-f028:**
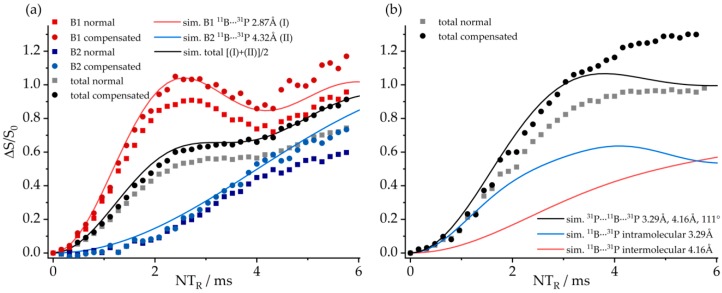
Individual ^11^B{^31^P} REDOR (squares) and compensated REDOR (circles) curves (see Figure 11) obtained for the integrated signal intensity of (**a**) **8** and (**b**) **7**. In the case of **8**, solid curves are two-spin simulations based on the crystal structure with ^11^B^…^^31^P distances of 2.87 Å (red curve, B1) and 4.32 Å (blue curve, B2). Black data points and solid curves show the weighted sum obtained if the total signal intensity is analyzed. In the case of **7**, simulations are shown for two-spin systems of intramolecularly and intermolecularly interacting ^11^B and ^31^P nuclei with internuclear distances of 3.29 Å (blue curve) and 4.16 Å (red curve). The black curve shows a three-spin simulation based on both distances and a ^31^P^…11^B^…31^P angle of 111° reflecting the structural situation in the crystal structure.

**Figure 29 molecules-25-01400-f029:**
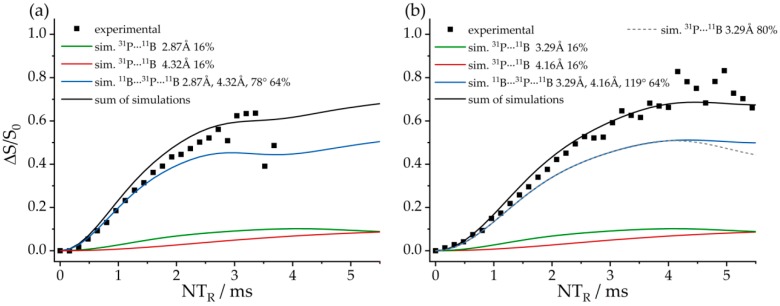
Experimental and simulated ^31^P {^11^B} REAPDOR curves for (**a**) **8** and (**b**) **7**, including the simulated components of the corresponding isotopologues based on the dipolar geometries as indicated.

**Figure 30 molecules-25-01400-f030:**
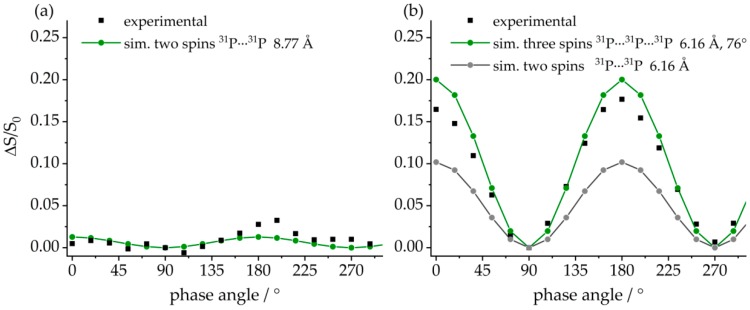
^31^P–^31^P CT-DRENAR curves (black squares) for (**a**) **8** and (**b**) **7** acquired for a dipolar recoupling time of 2.56 ms at 7.05 T, using a MAS frequency of 12.5 kHz with SWFTPPM [84] ^1^H decoupling. The simulation in (**a**) is based on a two-spin system with a ^31^P^…31^P distance of 8.77 Å (green curve) based on the shortest internuclear distance in the single-crystal structure. In (**b**), two simulations are depicted: a hypothetical dimer (two-spin system, gray curve) and the tetramer (three-spin system, green curve). Each simulation considered a ^31^P^…31^P distance of 6.16 Å and, in the latter case, a ^31^P^…31^P^…31^P angle of 76° based on the single-crystal structure. The agreement of the experimental data with the three-spin simulation gives clear evidence for the formation of the macrocycle.

**Figure 31 molecules-25-01400-f031:**
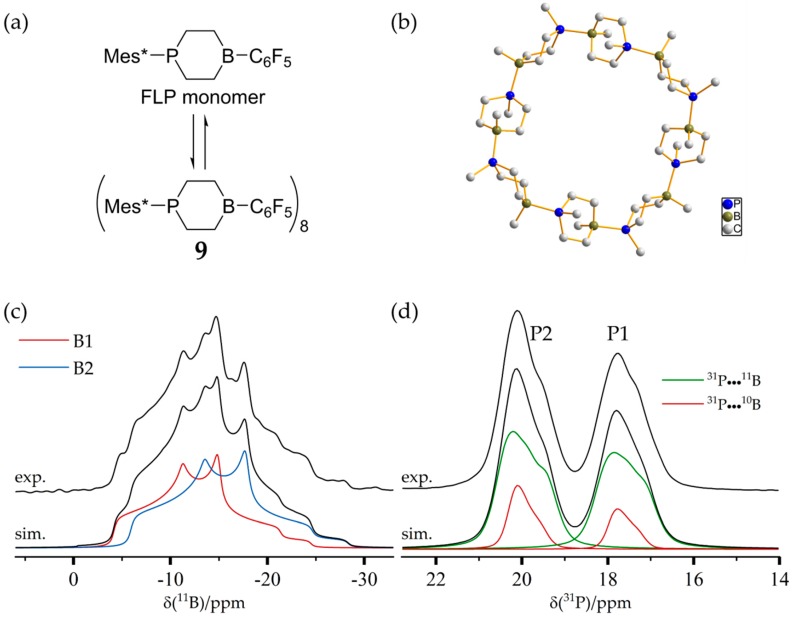
(**a**) Schematic representation of FLP monomer **6** and FLP-cyclooctamer **9** and (**b**) molecular structure of **9**, as well as the respective experimental and simulated (**c**) ^11^B and (**d**) ^31^P MAS NMR spectra of **9**.

**Figure 32 molecules-25-01400-f032:**
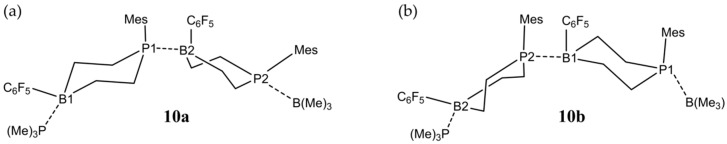
Chair- and boat-like conformational geometries present in **9**. To facilitate the DFT calculation of the NMR observables, the simplified molecules **10a** and **10b** shown served as models for the cutouts of the molecular structure of **9**, where, for instance, NMR parameters of P1 and B2 were calculated using **10a**.

**Figure 33 molecules-25-01400-f033:**
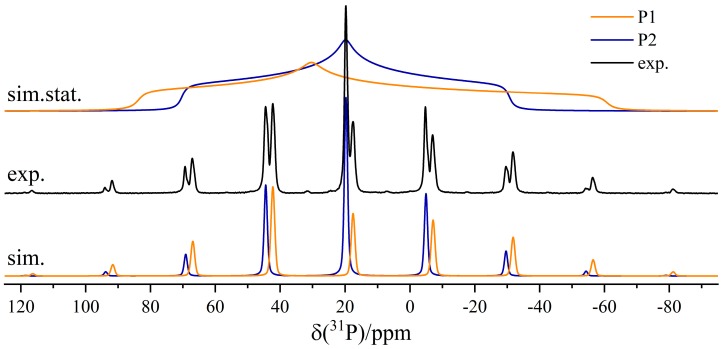
Experimental and simulated slow-spinning ^31^P MAS NMR spectra (spinning frequency 3 kHz) of the cyclooctamer **9**, revealing clear differences in the ^31^P chemical shift anisotropies of the two phosphorus sites. Simulations of the corresponding static ^31^P NMR spectra are included as well.

**Figure 34 molecules-25-01400-f034:**
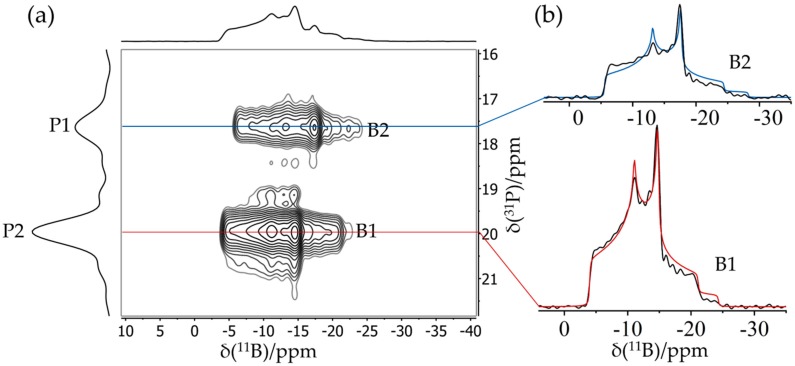
(**a**) 2D ^11^B {^31^P} CP/refocused INEPT experiment on **9**. The assignment of boron species results from the covalent bond of P1 in the chair conformation of the six-membered ring with B2 in the boat conformation and vice versa for P2 and B1. (**b**) Horizontal slices of each species (black line) and ^11^B line-shape simulations (blue and red curves).

**Figure 35 molecules-25-01400-f035:**
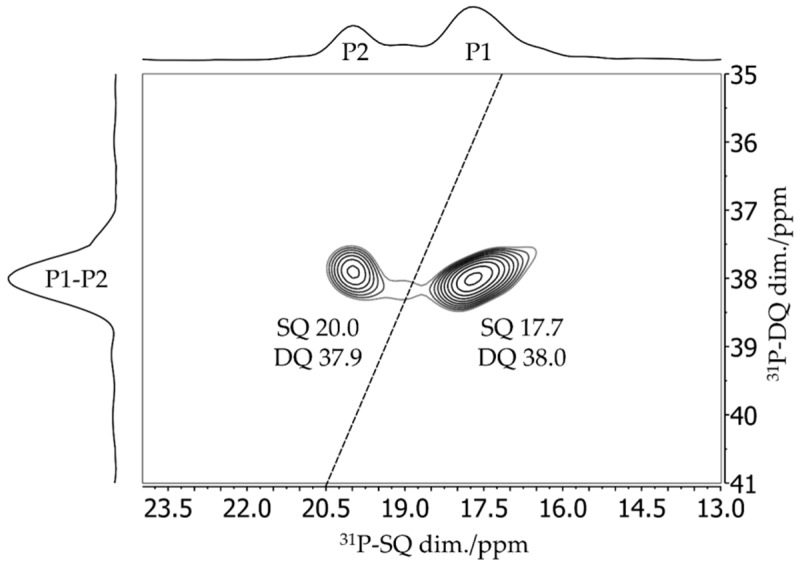
The 2D ^31^P{^1^H} CP/BaBa NMR spectrum of **9**, acquired with four BaBa blocks with four 90° pulses each (see Figure 10). ^31^P chemical shifts in the single-quantum (SQ) and DQ dimensions are assigned to the resonances revealing P1–P2 correlation peaks, whereas no P1–P1 or P2–P2 autocorrelations are observed.

**Figure 36 molecules-25-01400-f036:**
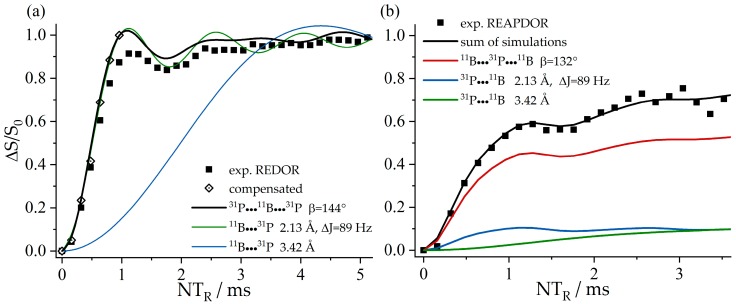
(**a**) Experimental and simulated ^11^B{^31^P} REDOR curves of **9**. Two-spin simulations based on the shortest intermolecular distance, including a calculated *J-*anisotropy of 89 Hz (green curve) and the intramolecular distance (blue curve). Black curve: Three-spin simulation based on the two closest boron–phosphorus distances in the specific dipolar geometry present in the crystal structure. (**b**) ^31^P{^11^B} REAPDOR curves for **9** showing simulations for the three distinct isotopologue contributions to the overall response (black) curve.

**Figure 37 molecules-25-01400-f037:**
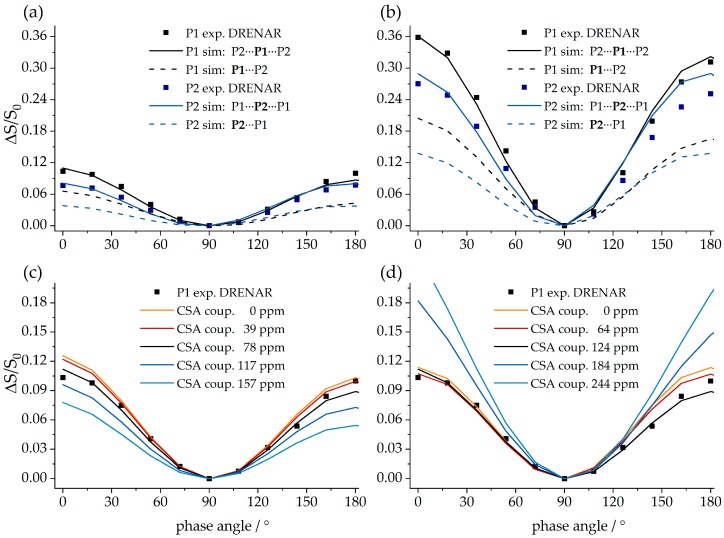
^31^P CT-DRENAR curves for species P1 (black squares) and P2 (blue squares) of the octameric compound **9** acquired using a dipolar mixing time of (**a**, **c**, **d**) 1.28 ms and (**b**) 2.56 ms at 7.05 T with a MAS spinning frequency of 12.5 kHz and SWFTPPM ^1^H decoupling. In (**a**) and (**b**), simulations for each species based on a two-spin (dashed curves) and a three-spin system (solid curves) are included based on a ^31^P^…31^P distance of 5.30 Å. In addition, experimentally determined ^31^P CSAs were considered in all simulations, as well as the angles between dipole–dipole and CSA interaction tensors from the single-crystal structure. The agreement of the experimental data with the three-spin simulations gives evidence for the formation of the octameric macrocycle. Additionally, simulations for P1 using various ^31^P CSAs for (**c**) the coupling nuclei and (**d**) the observed nucleus are depicted to demonstrate the ^31^P CSA influence on the ^31^P–^31^P CT-DRENAR curve.

**Table 1 molecules-25-01400-t001:** Solid-state NMR parameters of FLP Aggregates **1**–**9**
^a^.

Comp.	^11^B Parameters				^31^P Parameters				J-Coupling	
	Species	$\delta_{CS}^{iso}\pm \;0.5\;(\mathrm{ppm})$	$C_{Q}\;\pm \;0.05\;(\mathrm{MHz})$	$\eta_{Q}\;$ ± 0.05	Coord. ^31^P species	$\delta_{CS}^{iso}\pm \;0.5\;(\mathrm{ppm})$	Δσ± 5 (ppm)	*η* _σ_ ± 0.05	$J{(}^{31}P{-}^{11}B)\;\pm \;5\;(\mathrm{Hz})\;$	Calc. Δ*J* (Hz)
**1**	B	−4.6	1.21	0.63	P	48.9	93.8	0.83	49	
		−4.2	1.34	0.62		57.0	120.1	0.74	41/22	
**2**	B	−2.1	0.84	0.48	P	11.7	49.5	0.43		
		−4.6	0.94	0.42		5.8	55.4	0.40	13/17	
**3**	B	−14.6	1.69	0.35	P	17.4	99	0.57		
		−17.6	1.67	0.36		26.4			61.8	92
**4**	B1	−9.5	1.10	0.20	P1	21.8			66	
		−11.3	1.11	0.25		26.6			62	
	B2, B4	8.2	2.30	0.30						
		6.7/5.8	2.32/2.33	0.30						
	B3	−9.3	1.00	0.20	P2	20.8			70	
		−11.3	1.03	0.26		22.3			70	
**5**	B1	−9.5	1.08	0.30	P1	14.8			80	
		−10.9	1.17	0.11		18.0			69	92
	B2	7.3	2.38	0.25						
		7.5	2.49	0.26						
	B3	−8.3	1.05	0.30	P2	13.2			82	
		−12.0	1.04	0.17		14.1			78	92
	B4	6.5	2.37	0.28						
		5.92	2.27	0.33						
	B5	−8.9	1.10	0.30	P3	11.7			78	
		−13.0	1.13	0.06		12.7			75	92
	B6	6.9	2.36	0.28						
		6.13	2.24	0.33						
	B7	−8.9	1.09	0.20	P4	10.1			70	

	B8	7.5	2.21	0.15						

**7**	B1A	1.3	2.17	0.50	P	14.6	90	0.42		
		1.0	2.30	0.52						
	B1B	1.2	2.04	0.47		15.5	101	0.41	13	
		−1.2	2.18	0.52						
**8**	B1	5.8	2.69	0.24						
		3.8	2.78	0.17						
	B2	0.7	2.13	0.05	P	3.9	43	0.95	42	
		−0.8	2.23	0.10		−2.8	56	0.38	41	
**9**	B1	−3.7	1.70	0.64	P2	20.0	78.4	0.96	38	
		−5.0	1.79	0.57		24.5	70.0	0.86	33	89
	B2	−5.4	1.79	0.62	P1	17.7	123.8	0.68	40	
		−4.0	1.90	0.72		24.7	128.0	0.74	28	89

^a^ DFT calculated values are highlighted in gray. Distinct sites are identified according to their crystallographic information.

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
