# Peer review of "Solid-State NMR Techniques for the Structural Characterization of Cyclic Aggregates Based on Borane–Phosphane Frustrated Lewis Pairs"

_molecules, 2020, doi:10.3390/molecules25061400_

Round 1

Reviewer 1 Report

I note at the outset that I am not evaluating the science reported in this review, since all of it has already passed peer review, but only the value added by compiling it into a review format for this particular journal's audience.

This manuscript is a review of the authors' recent work using NMR to study the aggregation behaviour of borane-phosphane adducts in the solid state. They have published similar reviews of their work in this area in the past; this one focuses on the last five years. It is well written and informative, featuring high-quality experimental data and authoritative interpretations leading to chemically valuable conclusions. Unique to this work is the sophistication of the NMR techniques brought to bear upon the structural questions, relying on effective collaborations with trustworthy synthetic and computational chemists.

However, it is hard to know who is the intended audience, since this manuscript focuses tightly upon the work of one group, serving as little more than an illustrated and annotated guide to six recent papers; surely those who study these systems will already be familiar with these papers. Likewise, the lengthy introduction to NMR is paradoxically both too detailed for the non-specialist, and too superficial for the practitioner. It is admittedly a difficult challenge to find the right balance in a background theory section, but I am not sure they have found it. I would speculate that this review is meant to draw the attention of NMR spectroscopists to work published in general chemistry journals, which may have escaped notice.

One overarching criticism of this work lies in its framing. The introductory section justifying the use of NMR in the study of BP FLPs is well crafted, highlighting the limitations of other techniques and how NMR is uniquely suited to provide key structural information in poorly crystalline and disordered solids. However, the rest of the manuscript deals exclusively with cases where crystal structures have been independently determined, undermining the appropriateness of the introduction. Specifically, it is hard to reconcile the last sentence of the conclusion (848ff) and the last sentences of the introduction (74ff) with the material that is actually presented in the body of the text, with the possible exception of an oblique reference to "important structural constraints" (483) placed on a polymeric material, apparently the subject of an unpublished Ph.D. thesis. These assertions would be more convincing if examples of unsolved structures or amorphous solids were presented.

Despite these misgivings, I would not want to stand in the way of its publication, particularly as it would appear in a special issue whose readership would most likely find parts of it engaging and valuable. I recommend publication after these and the following points are considered by the authors.

  1. The document I received was missing figures 3 and 25, the figure captions positioned below blank spaces. Nevertheless, I doubt my evaluation of the manuscript would be altered by seeing them.
  2. One of the strengths of the extensive NMR theory section is the referencing, which will lead readers to more comprehensive explanations of key points. However, there are a few places were additional references are called for: 127, 256, 273, 287. The last few lines of the computational details would also benefit from better referencing (875-878). Also, a specific reference to the "exemplary" spectrum in line 419 would be helpful.
  3. There is some ambiguity about how the J-anisotropy was determined. Since it has the same orientational dependence as R, it is only possible to determine if the internuclear distance is known, whence the "effective R" may differ from that calculated from the crystal structure. Alternately, the value may be calculated from first principles. The authors should be more clear about how the reported values were obtained, both in the text and the table.
  4. Reference 55 in line 310 appears to be incorrect. And it is out of order.
  5. Lines 155-156 should be omitted as they are redundant with the statements in the following section.
  6. Is the middle C missing a H in line 508...?
  7. The experimental data points should be labelled in figure 15.

Author Response

Responses to Referee I.

Comment

This manuscript is a review of the authors' recent work using NMR to study the aggregation behaviour of borane-phosphane adducts in the solid state. They have published similar reviews of their work in this area in the past; this one focuses on the last five years. It is well written and informative, featuring high-quality experimental data and authoritative interpretations leading to chemically valuable conclusions. Unique to this work is the sophistication of the NMR techniques brought to bear upon the structural questions, relying on effective collaborations with trustworthy synthetic and computational chemists.

Response:

We appreciate the positive comments of the referees

 Comment:

However, it is hard to know who is the intended audience, since this manuscript focuses tightly upon the work of one group, serving as little more than an illustrated and annotated guide to six recent papers; surely those who study these systems will already be familiar with these papers. Likewise, the lengthy introduction to NMR is paradoxically both too detailed for the non-specialist, and too superficial for the practitioner. It is admittedly a difficult challenge to find the right balance in a background theory section, but I am not sure they have found it. I would speculate that this review is meant to draw the attention of NMR spectroscopists to work published in general chemistry journals, which may have escaped notice.

Response:

We agree with the referee that striking the right balance in review articles on subjects for a general audience is a challenge. The intention of this review is not to write an article for NMR specialists who may have missed papers in general chemistry, but rather to draw attention to materials chemists, who are familiar with solid-state NMR at some level, towards the possibilities of advanced NMR techniques beyond simple single-pulse or cross-polarization magic angle spinning experiments. This requires some theoretical background for explaining those techniques, which we intended to present in a non-mathematical fashion. In the revised version we added some more information that may be useful for the practitioner.

Comment

One overarching criticism of this work lies in its framing. The introductory section justifying the use of NMR in the study of BP FLPs is well crafted, highlighting the limitations of other techniques and how NMR is uniquely suited to provide key structural information in poorly crystalline and disordered solids. However, the rest of the manuscript deals exclusively with cases where crystal structures have been independently determined, undermining the appropriateness of the introduction. Specifically, it is hard to reconcile the last sentence of the conclusion (848ff) and the last sentences of the introduction (74ff) with the material that is actually presented in the body of the text, with the possible exception of an oblique reference to "important structural constraints" (483) placed on a polymeric material, apparently the subject of an unpublished Ph.D. thesis. These assertions would be more convincing if examples of unsolved structures or amorphous solids were presented.

 Response:

This criticism is correct and understandable. We have modified the Introduction and the Conclusions with this critique in mind, de-emphasizing the use of solid state NMR for clarifying structural issues in disordered or heterogeneous systems, as indeed the body of the text does not contain such examples. The purpose of the present contribution is to lay the foundations, by focusing on NMR measurements of crystallographically well-characterized materials. (Our previous work on FLP polymers was reviewed more extensively in reference 19, so there is no need for repetition). Other research on disordered and heterogeneously organized FLPs is at very initial stages. In the Conclusion Section we are now citing a specific recent example from a different group, which may benefit from the application of more advanced NMR methodology. Finally, beyond our article’s focus on FLPs, we hope to have shown how advanced solid-state NMR strategies can be used for the structural characterization of supramolecular aggregates in other areas of materials chemistry.

Comment:

Despite these misgivings, I would not want to stand in the way of its publication, particularly as it would appear in a special issue whose readership would most likely find parts of it engaging and valuable. I recommend publication after these and the following points are considered by the authors.

  1. The document I received was missing figures 3 and 25, the figure captions positioned below blank spaces. Nevertheless, I doubt my evaluation of the manuscript would be altered by seeing them.

Response:

Hopefully the figures convert satisfactorily in this new submission.

Comment:

  1. One of the strengths of the extensive NMR theory section is the referencing, which will lead readers to more comprehensive explanations of key points. However, there are a few places were additional references are called for: 127, 256, 273, 287. The last few lines of the computational details would also benefit from better referencing (875-878). Also, a specific reference to the "exemplary" spectrum in line 419 would be helpful.

Response:

Following the referee’s suggestion, we have amplified the referencing in various appropriate places. The spectrum in line 419 is found in reference [79], which is publicly available.

Comment:

  1. There is some ambiguity about how the J-anisotropy was determined. Since it has the same orientational dependence as R, it is only possible to determine if the internuclear distance is known, whence the "effective R" may differ from that calculated from the crystal structure. Alternately, the value may be calculated from first principles. The authors should be more clear about how the reported values were obtained, both in the text and the table.

Response:

Actually, we used both approaches for determining the J-anisotropy. If not specified, reported values are based on the lineshape simulation fitted to the experimental data, based on the known dipolar coupling constant calculated from the crystal structure. In addition, these values were also determined by ab-initio calculations and are reported as such accordingly. Both approaches were generally found in good agreement. We have clarified this point in the text, lines 560-563.

Comment

  1. Reference 55 in line 310 appears to be incorrect. And it is out of order.

Response

Thank you very much. We have corrected this error

Comment:

  1. Lines 155-156 should be omitted as they are redundant with the statements in the following section.

Response:

We agree this needed correction and have replaced “isotropic average” by “isotropic value”. .

Comment:

  1. Is the middle C missing a H in line 508...?

Response:

Correction done

Comment:

  1. The experimental data points should be labelled in figure 15.

Response:

We have clarified this point by re-writing the figure caption of this Figure (now Figure 17).

Reviewer 2 Report

This manuscript is the excellent review which describes solid-state NMR analysis of frustrated Lewis pairs. FLP forms nano- to microcrystals, amorphous, and aggregates with possible structural disorders. While widely used single crystal XRD analysis fails to determine molecular structures for these FLP samples, ssNMR provides fruitful information on atomic level structures. In this manuscript, first various ssNMR techniques used in proving 11B and 31P nuclei in B/P FLP samples are introduced. This includes introduction of spin interactions and its experimental determination by ssNMR techniques. Then, the experimental demonstrations on FLP are introduced. The manuscript is very well written and covers the recent progress of ssNMR analysis of FLP. I recommend this article for publication in molecules after addressing the following questions.

I think MQMAS together with its experimental demonstrations (or introduction of previous literatures) should be included in the manuscript. Although the manuscript introduces most of ssNMR techniques, MQMAS is not described at all. MQMAS is key techniques to separate overlapped 11B resonances and to determine quadrupolar coupling precisely. For example, ref 20 (JACS 124 (2012) 4236) extensively used MQMAS.

minor question.

Page 5 line 150: The equations can be confusing. These can be replaced with sigma_iso = (s33+s22+s11)/3 and delta_sigma = s33 – (s11+s22)/2

Page 11 figure 6(a): Why 90 degree pulse is applied on S just before acquisition?

Page 11 line 295: What does the isotropic indirect dipolar interaction stand for? Residual dipolar splitting coming from cross term between quadrupolar interaction and heteronuclear dipolar interaction?

Author Response

Responses to Referee II

Comment

This manuscript is the excellent review which describes solid-state NMR analysis of frustrated Lewis pairs. FLP forms nano- to microcrystals, amorphous, and aggregates with possible structural disorders. While widely used single crystal XRD analysis fails to determine molecular structures for these FLP samples, ssNMR provides fruitful information on atomic level structures. In this manuscript, first various ssNMR techniques used in proving 11B and 31P nuclei in B/P FLP samples are introduced. This includes introduction of spin interactions and its experimental determination by ssNMR techniques. Then, the experimental demonstrations on FLP are introduced. The manuscript is very well written and covers the recent progress of ssNMR analysis of FLP. I recommend this article for publication in molecules after addressing the following questions.

Response:

We appreciate the positive comments of the referee.

Comment:

I think MQMAS together with its experimental demonstrations (or introduction of previous literatures) should be included in the manuscript. Although the manuscript introduces most of ssNMR techniques, MQMAS is not described at all. MQMAS is key techniques to separate overlapped 11B resonances and to determine quadrupolar coupling precisely. For example, ref 20 (JACS 124 (2012) 4236) extensively used MQMAS.

Response:

We agree with the referee and have included the topic of MQMAS in the manuscript. We have further added an application figure documenting its ability of separating heavily overlapping resonances in one application example. Two new figures (Figure 5 and 6) have been added to the manuscript.

Comment:

minor question.

Page 5 line 150: The equations can be confusing. These can be replaced with sigma_iso = (s33+s22+s11)/3 and delta_sigma = s33 – (s11+s22)/2

Response:

The equations are shown within the text on lines 147-148.

Comment:

Page 11 figure 6(a): Why 90 degree pulse is applied on S just before acquisition?

Response: in our understanding the purpose of this pulse is to eliminate any transverse S-magnetization created by the foregoing 90° pulse that stops the double-quantum coherence evolution. As only I-magnetization is read out, the presence of that final 90°(S) pulse is probably optional.

Comment:

Page 11 line 295: What does the isotropic indirect dipolar interaction stand for? Residual dipolar splitting coming from cross term between quadrupolar interaction and heteronuclear dipolar intebbbraction?

Response:

The referee is correct. We have modified the figure caption accordingly.